# Deep Geodesic Canonical Correlation Analysis for Covariance-Based Neuroimaging Data

**Ce Ju**[1,†]  **Reinmar J. Kobler**[2,3,†]  **Liyao Tang**[4]  **Cuntai Guan**[1]  **Motoaki Kawanabe**[2,3]

[1] School of Computer Science and Engineering, Nanyang Technological University, Singapore
[2] Advanced Telecommunications Research Institute International, Japan
[3] RIKEN Artificial Intelligence Project, Japan
[4] School of Computer Science, University of Sydney, Australia
[†] Equal contribution

## Abstract

In human neuroimaging, multi-modal imaging techniques are frequently combined to enhance our comprehension of whole-brain dynamics and improve diagnosis in clinical practice. Modalities like electroencephalography and functional magnetic resonance imaging provide distinct views to the brain dynamics due to diametral spatiotemporal sensitivities and underlying neurophysiological coupling mechanisms. These distinct views pose a considerable challenge to learning a shared representation space, especially when dealing with covariance-based data characterized by their geometric structure. To capitalize on the geometric structure, we introduce a measure called geodesic correlation which expands traditional correlation consistency to covariance-based data on the symmetric positive definite (SPD) manifold. This measure is derived from classical canonical correlation analysis and serves to evaluate the consistency of latent representations obtained from paired views. For multi-view, self-supervised learning where one or both latent views are SPD we propose an innovative geometric deep learning framework termed DeepGeoCCA. Its primary objective is to enhance the geodesic correlation of unlabeled, paired data, thereby generating novel representations while retaining the geometric structures. In simulations and experiments with multi-view and multi-modal human neuroimaging data, we find that DeepGeoCCA learns latent representations with high geodesic correlation for unseen data while retaining relevant information for downstream tasks.

## 1    Introduction

In human neuroimaging, it is common to use multi-modal imaging techniques to improve our understanding of the dynamic relationships among brain networks and their alterations in pathology (Debener et al., 2006; Ou et al., 2010; Huster et al., 2012; Sadaghiani & Wirsich, 2020). For instance, combining non-invasive electroencephalography (EEG) measurements with functional magnetic resonance imaging (fMRI) can offer a high-resolution view of neurophysiological events relevant to treat mental disorders like epilepsy (Moeller et al., 2009). However, achieving a sense of *consistency* between EEG and fMRI frequently presents significant challenges due to a range of complications stemming from their disparities (Daunizeau et al., 2009; Warbrick, 2022).

A classical statistical learning approach to identify paired subspaces for consistency through correlation is Canonical Correlation Analysis (CCA) (Hotelling, 1992). Its versatility is evident in a range of scenarios even when the sample size is limited relative to data dimensionality or when data dimensionality exceeds human interpretability (Uurtio et al., 2017). Beyond classical CCA, extensions to study phase-amplitude coupling and amplitude-amplitude coupling proved useful in uni-modal (Dähne et al., 2014) and multi-modal neuroimaging (Dähne et al., 2015; Deligianni et al., 2014). For instance, Deligianni et al. (2014; 2016) used inverse modeling to align paired, resting-state EEG and fMRI signals spatially. After alignment they used a CCA variant to link the covariance-based EEG and fMRI data, and assessed the goodness-of-fit with a geometry-aware evaluation criterion.

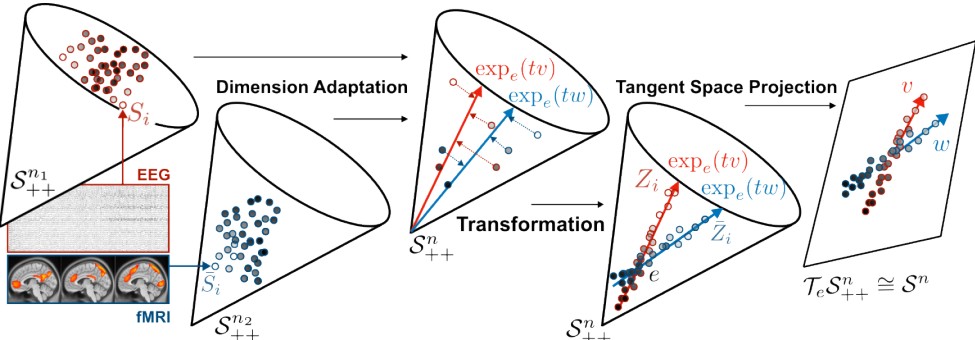

Figure 1: **Diagram of DeepGeoCCA Framework**. We consider paired views of multivariate time-series data (e.g., simultaneously recorded EEG and fMRI data). Individual views are transformed (e.g., covariance matrices) to form $N$ paired views on SPD manifolds with different dimensions $n_1$ and $n_2$, i.e., $\{(S_i, \bar{S}_i) \mid S_i \in \mathcal{S}_{++}^{n_1}, \bar{S}_i \in \mathcal{S}_{++}^{n_2}\}_{i=1}^{N}$. A Dimension Adaptation module transforms the views so that they lie on $\mathcal{S}_{++}^{n}$, a cone manifold in $\mathbb{R}^{n \times n}$. On this cone, a Transformation module learns to center the data around the identify matrix $e$ and project them onto geodesics $\exp_e(tv)$ and $\exp_e(tw)$ to form latent SPD representations $\{(Z_i, \bar{Z}_i)\}_{i=1}^{N}$. The aim of this problem is to maximize geodesic correlation which measures correlation along the tangent speeds $v$ and $w$, after the Tangent Space Projection at $e$.

When dealing with paired covariance-based neuroimaging data, using conventional CCA methods may lead to a loss of certain mathematical structures that are important for practical results, such as properties of symmetry and positive definiteness (SPD). Modeling techniques based on Riemannian geometry have become the primary approach to address such constraints in covariance-based neuroimaging data (Fillard et al., 2007; Barachant et al., 2011; Qiu et al., 2015; You & Park, 2021; Dan et al., 2022). In the case of CCA, extensions generalized the concept of subspace projection in Euclidean vector space to non-linear projections on geodesic submanifolds for general Riemannian manifolds (Kim et al., 2014) and SPD manifolds (Kim et al., 2014; Fallah & Yang, 2020).

In the context of multi-view deep representation learning (Wang et al., 2015), CCA inspired a family of self-supervised learning (SSL) approaches (Caron et al., 2020; Zbontar et al., 2021; Ermolov et al., 2021; Bardes et al., 2022), as categorized in Balestriero et al. (2023). Methods, such as Deep-CCA (Andrew et al., 2013) and VICReg (Bardes et al., 2022), proposed to learn the correlation relationship between two latent representations by analyzing their covariance and cross-covariance structure. However, for covariance-based neuroimaging data with specific geometric characteristics, these methods fail to maintain these particular geometric properties when learning correlation relationships.

Here, we propose a geometric deep learning framework, named Deep Geodesic Canonical Correlation Analysis, or DeepGeoCCA. It belongs to the SSL-based deep representation learning family (Chen et al., 2020) as well as geometric modeling techniques to learn latent SPD representations with high correlation consistency among paired views (Figure 1). To characterize this correlation consistency, we present a new correlation measure called geodesic correlation, which is applied to SPD manifolds equipped with the affine-invariant Riemannian metric (AIRM) (Pennec et al., 2006). Intuitively, geodesic correlation essentially measures correlation of paired data along geodesics passing through the identity matrix $e$. To maximize this measure, DeepGeoCCA introduces a novel geodesic loss function and SPD matrix-valued neural network models to learn latent SPD representations. Specifically, we devise a relaxation method, called $\varepsilon-$geodesic constraint, that enables deep neural networks on SPD manifolds (Huang & Gool, 2017; Ju & Guan, 2022; 2023; Kobler et al., 2022a) to map covariance-based neuroimaging data onto or near geodesics within a double-cone region around $e$.

Using theoretical analysis and experiments with simulations and real data, we systematically study our framework, detailed in section 3, and show that DeepGeoCCA:

- generates latent representations which maintain the properties of symmetry and positive definiteness.
- supports a *controllable* projection region. Specifically, we prove that adjusting $\varepsilon$ in our relaxation approach can control the deviation of the mapped points from the geodesics.

- allows *flexible* unit speeds for predefined geodesics, in the sense that they can be pre-defined by characteristics of downstream tasks or driven by data.
- effectively learns latent representations exhibiting significant geodesic correlation for previously unseen data while preserving pertinent information (section 4).

## 2 PRELIMINARY

We first briefly recap standard CCA, a deep learning-based CCA extension, and a CCA method on general Riemannian manifolds. Please refer to Appendix A for introducing SPD manifolds equipped with the affine-invariant Riemannian metric, Appendix B for Fréchet means and variances, Appendix C for geodesics. We extensively examine and contrast the following CCA variants in Appendix D.

### 2.1 NOTATIONS AND CONVENTIONS

In the following context, we employ the set $\mathcal{S}_{++}^n := \{S \in \mathbb{R}^{n \times n} : S = S^\top \text{ and } x^\top S x > 0, \forall x \in \mathbb{R}^n / \mathbf{0}\}$ to be the space of real $n \times n$ symmetric and positive definite matrices. The set $\mathcal{S}^n := \{S \in \mathbb{R}^{n \times n} | S = S^\top\}$ is the space of $n \times n$ real symmetric matrices. We always use $e$ [1] to represent the $n \times n$ identity matrix $I_n \in \mathcal{S}_{++}^n$. $(\mathcal{S}_{++}^n, g^{AIRM})$ to refer to the Riemannian manifold formed by equipping $\mathcal{S}_{++}^n$ with the AIRM. $\|\cdot\|_e$ is the Riemannian norm at identity $e$. The notation $\mathcal{F}$ denotes the Frobenius norm. $\langle \cdot, \cdot \rangle_{\ell^2}$ represents the $\ell^2-$inner product. $\exp(\cdot)$ and $\log(\cdot)$ are the matrix exponential and the matrix logarithm. $\text{Tr}(\cdot)$ is the trace of a matrix. The operator $vec(\cdot)$ represents the vectorization of a matrix.

### 2.2 CANONICAL CORRELATION ANALYSIS (HOTELLING, 1992)

Consider two paired matrices of data distributions $X \in \mathbb{R}^{N \times p}$ and $\bar{X} \in \mathbb{R}^{N \times q}$, where $N$ represents the number of observations, while $p$ and $q$ denote the dimensions of two variables. *Paired* means that each $i^{th}$ row in both $X$ and $\bar{X}$ are coupled (e.g., recorded at the same time). We assume that the columns in both $X$ and $\bar{X}$ are a zero-mean. The aim of CCA is to find two linear transformations, denoted as $w \in \mathbb{R}^p$ and $\bar{w} \in \mathbb{R}^q$, in such a way that the cosine of the angle $\Theta$ between the projected vectors $z^\dagger$ and $\bar{z}^\dagger$ onto the unit ball is maximized, as described below, $\cos \Theta := \max_{z^\dagger, \bar{z}^\dagger \in \mathbb{R}^N} \langle z^\dagger, \bar{z}^\dagger \rangle$, where $z^\dagger := (X \cdot w)/\|X \cdot w\| \in \mathbb{R}^N$, and $\bar{z}^\dagger := (\bar{X} \cdot \bar{w})/\|\bar{X} \cdot \bar{w}\| \in \mathbb{R}^N$.

### 2.3 DEEP CANONICAL CORRELATION ANALYSIS (ANDREW ET AL., 2013)

Deep canonical correlation analysis (DeepCCA) employs neural networks to learn latent representations with the goal maximize the CCA objective between latent representations of two modalities, as follows, $\max_{\theta, \bar{\theta}} \text{corr}\big(\varphi(x; \theta), \bar{\varphi}(\bar{x}; \bar{\theta})\big)$, where $\varphi(\cdot; \theta)$ and $\bar{\varphi}(\cdot; \bar{\theta})$ are neural networks with parameters $\theta$ and $\bar{\theta}$, respectively. Inspired by CCA, Andrew et al. (2013) proposed to compute and maximize $\left\|\sum_{\varphi\varphi}^{-1/2} \cdot \sum_{\varphi\bar{\varphi}} \cdot \sum_{\bar{\varphi}\bar{\varphi}}^{-1/2}\right\|_{\mathcal{F}}$ for each batch, where $\sum_{\varphi\varphi}$ and $\sum_{\bar{\varphi}\bar{\varphi}}$ represent within-modality covariances of latent representation, and $\sum_{\varphi\bar{\varphi}}$ represents cross-modality covariances of latent representations. Their preprocessing procedure includes centering and regularization.

### 2.4 RIEMANNIAN CANONICAL CORRELATION ANALYSIS (KIM ET AL., 2014)

Given a set of paired manifold-valued data $\{x_i\}_{i=1}^N$ and $\{\bar{x}_i\}_{i=1}^N \in (\mathcal{M}, g)$, Riemannian canonical correlation analysis (RieCCA) is the correlation between orthogonal projections of $\{x_i\}_{i=1}^N$ and $\{\bar{x}_i\}_{i=1}^N$ onto two geodesic submanifolds $\exp_\mu(U), \exp_{\bar{\mu}}(\bar{U}) \subset \mathcal{M}$, where two open neighborhoods $U \subset \mathcal{T}_\mu \mathcal{M}$ and $\bar{U} \subset \mathcal{T}_{\bar{\mu}} \mathcal{M}$ are spanned by tangent vectors at Fréchet means $\mu$ and $\bar{\mu}$, respectively, and two orthogonal projections $x_i^\dagger := \min_{x \in \exp_\mu(U)} d_g^2(x_i, x)$ and $\bar{x}_i^\dagger := \min_{\bar{x} \in \exp_{\bar{\mu}}(\bar{U})} d_g^2(\bar{x}_i, \bar{x})$ are on the geodesic submanifolds, for all $i \in \{1, ..., N\}$, where $d_g(\cdot, \cdot)$ is the Riemannian distance with respect to the Riemannian metric $g$.

---

[1] $e$ is the symbol that is typically utilized as the identity element in Lie group and Lie algebra contexts.

## 3 Deep Geodesic CCA (DeepGeoCCA) Framework

Consider paired views of covariance-based neuroimaging data, whether from the same or different modalities, using established criteria. For example, EEG and fMRI views are paired based on their recording timestamp. In the proposed framework, the need for labels is eliminated, offering convenience for neuroimaging data derived from complex brain dynamics. Deep neural networks on SPD manifolds are utilized to learn latent SPD matrix-valued representations by maximizing geodesic correlation. As such, DeepGeoCCA can be considered as a specific instance within the SSL family, designed particularly for handling paired SPD matrix-valued representations.

Without loss of generality, suppose $N$ paired SPD matrix-valued representations $\{S_i\}_{i=1}^N$ and $\{\bar{S}_i\}_{i=1}^N$ are on $(\mathcal{S}_{++}^n, g^{AIRM})$, each with Fréchet mean $e$, and two arc-length geodesics $\exp_e(tv)$ and $\exp_e(tw) : \mathcal{S}^n \longmapsto \mathcal{S}_{++}^n$ with time $t \in \mathbb{R}$ and tangent unit speed $v, w \in \mathcal{T}_e\mathcal{S}_{++}^n \cong \mathcal{S}^n$, respectively. Geodesic correlation is defined as follows with an illustration in Figure 2:

**Definition** (Geodesic Correlation). *Geodesic correlation with respect to the given tangent speeds $v$ and $w$ on $(\mathcal{S}_{++}^n, g^{AIRM})$ is defined as the correlation between their Riemannian logarithm of the orthogonal projections on the tangent space at the identity matrix $e \in \mathcal{S}_{++}^n$ as follows,*

$$corr(\{S_i\}_{i=1}^N, \{\bar{S}_i\}_{i=1}^N) := \frac{\sum_{i=1}^N g_e^{AIRM}\left(\log_e(S_i^\dagger), \log_e(\bar{S}_i^\dagger)\right)}{\sqrt{\sum_{i=1}^N \left\|\log_e(S_i^\dagger)\right\|_e^2} \cdot \sqrt{\sum_{i=1}^N \left\|\log_e(\bar{S}_i^\dagger)\right\|_e^2}}, \quad (1)$$

*where $S_i^\dagger$ and $\bar{S}_i^\dagger$ represent the orthogonal projections on the given geodesics.*

Note that all the orthogonal projections are unique, which is a result of the affine-invariant Riemannian metric that leads the SPD space to be a Cartan-Hadamard manifold (Bacák, 2014; Pennec, 2020). Hence, the geodesic correlation is *well-defined*. Please refer to Appendix C for orthogonal projection and geodesic constraints, Appendix E for additional motivations and main difference with Riemannian Correlation. Besides, incorporating geodesic correlation and introducing geodesic constraints can be reformulated as a constrained non-linear optimization problem, and its Lagrangian duality naturally serves as a loss function for the neural network-based solutions, as explained in Appendix F.

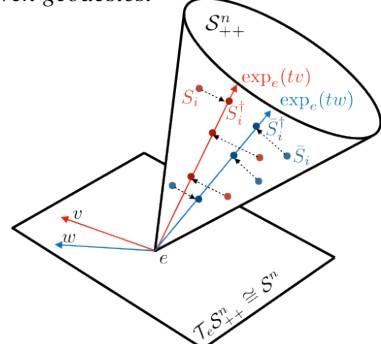

Figure 2: Geodesic Correlation.

### 3.1 Neural Network-Based Solution

As outlined in Figure 1, we transform paired views of covariance-based neuroimaging data into latent SPD matrix-valued representations centered around the identity matrix $e$. To that end, we employ neural network architectures on SPD manifolds. These specialized architectures can maintain the properties of symmetry and positive definiteness throughout their computations.

The proposed framework comprises three modules. The first module, known as *Dimension Adaptation* network, is responsible for converting input data of varying dimensions to a consistent dimension. The second module, referred to as *Transformation* network, learns non-linear transformations to center the data around the identity matrix $e$ and project them close to geodesics passing through $e$. Lastly, *Tangent Space Projection* maps the observations to the tangent space at $e$ to form inputs for our proposed loss terms. In particular, to enable computations to be performed at $e$, we employ Riemannian Batch Normalization on $(\mathcal{S}_{++}^n, g^{AIRM})$ (Brooks et al., 2019) with specially tuned step momentum parameter to parallel transport Fréchet means of SPD data distribution to $e$ (Kobler et al., 2022a). For a more detailed overview of the employed SPD layers, please refer to Appendix G.

### 3.2 Relaxed Orthogonal Projection

In this section, we propose a relaxation method for the geodesic constraints of geodesic correlation, which is particularly suited for the neural network-based solution. In intuitive terms, this relaxation assesses whether the projection $Z \cong S^\dagger$ on tangent space $\mathcal{T}_e\mathcal{S}_{++}^n$, after being normalized, falls

within the double cone centered at identity $e$, with the unit geodesic speed $u$ as the symmetric line, by calculating the cosine value between it and $u$, defined as follows,

$$\cos(\Theta) := \frac{g_e^{AIRM}\big(\log_e(Z), u\big)}{\|\log_e(Z)\|_e \cdot \|u\|_e}. \tag{2}$$

A relaxation parameter $\varepsilon \in [0, 1]$ is introduced to control how much the projections can deviate from the geodesic by $\varepsilon \le |\cos(\Theta)|$. We refer to this new constraint as the $\varepsilon-$geodesic constraint. Furthermore, we provide an estimate of the discrepancies between projections precisely on the geodesic and those within the tubular neighborhood of the geodesic $\exp_e(tu)$ as follows,

**Theorem.** *Suppose $\{Z_i\}_{i=1}^N$ are a set of latent representations on $(\mathcal{S}_{++}^n, g^{AIRM})$, and a $\delta$-width tubular neighborhood of arc-length geodesic $\exp_e(tu)$ with unit speed $u$ is defined as $\{\exp_e(tu + v)|v \in (\mathcal{T}_e\mathcal{S}_{++}^n)^\perp, \|v\|_e \le \delta, t \in \mathbb{R}\}$, where $(\mathcal{T}_e\mathcal{S}_{++}^n)^\perp$ represents the orthogonal complement of $\mathcal{T}_e\mathcal{S}_{++}^n$. Then, the width $\delta$ of this tubular neighborhood is upper bounded by the following inequality,*

$$\delta \le \sqrt{\frac{1 - \varepsilon^2}{\varepsilon^2}} \max_{i=\{1,\dots,N\}} g_e^{AIRM}\big(\log_e(Z_i), u\big),$$

*where relaxation parameter $\varepsilon \in (0, 1]$ regulates the deviation of the projections from the geodesics by $\varepsilon \le |cos(\Theta)|$. In particular, when $\varepsilon \to 1$, we have $\delta \to 0$.*

For a comprehensive proof, please consult Appendix H.

### 3.3 LOSS FUNCTION

Within the proposed framework, SPD neural nets are trained to minimize an overall loss function $\mathcal{L}$ comprising a correlation loss term $\mathcal{L}_\rho$, relaxed $\varepsilon-$geodesic constraints $\mathcal{L}_\varepsilon$, and a variance-preserving loss $\mathcal{L}_\sigma$:

$$\mathcal{L} := \alpha_1 \cdot \mathcal{L}_\rho + \alpha_2 \cdot \mathcal{L}_\varepsilon + \alpha_3 \cdot \mathcal{L}_\sigma. \tag{3}$$

with term coefficients $\alpha_1, \alpha_2$, and $\alpha_3 \ge 0$. (Tips of choice for term coefficients refers to Remark 2.)

Specifically, the correlation loss $\mathcal{L}_\rho$ is utilized to maximize geometric correlation among paired views as follows,

$$\mathcal{L}_\rho := -\frac{\sum_{i=1}^N \text{Tr}\big(\log_e(Z_i) \cdot \log_e(\bar{Z}_i)\big)}{\sqrt{\sum_{i=1}^N \text{Tr}\big(\log_e^2(Z_i)\big)} \cdot \sqrt{\sum_{i=1}^N \text{Tr}\big(\log_e^2(\bar{Z}_i)\big)}},$$

where $Z_i$ and $\bar{Z}_i$ denote the latent SPD matrix-valued representations for two paired views, respectively, and symmetric matrices $\log_e(Z_i)$ and $\log_e(\bar{Z}_i)$ are their projections on the tangent space at $e$; The relaxed $\varepsilon-$geodesic constraints $\mathcal{L}_\varepsilon$ are utilized to penalize projections outside the double-cone regions as follows,

$$\mathcal{L}_\varepsilon := \sum_{i=1}^N \min\big\{\varepsilon, 1 - \frac{\big|\text{Tr}\big(\log_e(Z_i) \cdot v\big)\big|}{\|\log_e(Z_i)\|_e \cdot \|v\|_e}\big\} + \sum_{i=1}^N \min\big\{\varepsilon, 1 - \frac{\big|\text{Tr}\big(\log_e(\bar{Z}_i) \cdot w\big)\big|}{\|\log_e(\bar{Z}_i)\|_e \cdot \|w\|_e}\big\},$$

where geodesic speeds $v, w \in \mathcal{S}^n$ are predefined or learnable parameters, and $\varepsilon$ is a preset relaxation parameter; Furthermore, a variance-preserving loss $\mathcal{L}_\sigma$ proposed in (Bardes et al., 2022) is utilized to mitigate potential dimensional collapse, which is frequently encountered in the SSL framework (Jing et al., 2022) as follows,

$$\mathcal{L}_\sigma := \frac{1}{d} \cdot \sum_{i=1}^d \max\big\{0, 1 - \sqrt{\text{Var}\big(S(Z)^i\big) + \epsilon}\big\} + \frac{1}{d} \cdot \sum_{i=1}^d \max\big\{0, 1 - \sqrt{\text{Var}(S(\bar{Z})^i) + \epsilon}\big\},$$

where $d$ is the dimension of $vec(\log_e(Z))$, $S(Z)^i$ is the vector composed of values at the $i^{th}$ dimension of $\big(vec(\log_e(Z_1)), ..., vec(\log_e(Z_N))\big) \in \mathbb{R}^{N \times d}$, and $\epsilon$ is a scalar to prevent numerical instabilities, and $\epsilon = 10^{-4}$ in practice.

**Remark 1.** *Choice of Initial Geodesic Speeds: The selection of the initial geodesic speeds for two covariance-based modalities can be influenced by the specific requirements of downstream tasks or solely dictated by the data. For example, a choice can be the largest canonical correlation obtained from the conventional CCA as $vec(v), vec(w) := CCA(\{vec(S_i)\}_{i=1}^N, \{vec(\bar{S}_i)\}_{i=1}^N)$;*

**Remark 2.** *Choice of Loss Term Coefficients: Depending on the specific scenario and architecture, the coefficients $(\alpha_1, \alpha_2, \alpha_3)$ can be determined to give more importance to geodesic correlation or the constraints. By default, we set $\alpha_1 = 1, \alpha_2 = 0.25, \alpha_3 = 0$ to give more weight to $\mathcal{L}_\rho$. If a specific architecture tends to dimensional collapse, we enable the variance-preserving loss ($\alpha_3 > 0$);*

**Remark 3.** *Extending Overall Loss function on $\mathcal{S}^n$: Since all loss terms in the overall Loss (3) operate with tangent space projections $\log_e(Z_i), \log_e(\bar{Z}_i) \in \mathcal{T}_e \mathcal{S}^n_{++}$ corresponding to the space of symmetric matrices $\mathcal{S}^n$, the proposed loss can also be computed for any neural network generating latent representations in $\mathcal{S}^n$. For example, the output $x \in \mathbb{R}^{n(n+1)/2}$ of any standard neural network layer can be transformed to $\mathcal{S}^n$ via applying $\mathrm{upper}^{-1}$ where the norm-preserving map $\mathrm{upper}$ vectorizes a matrix's elements along the upper triangular part.*

## 4 EXPERIMENTS

We conducted simulations and two experiments with public human datasets to empirically evaluate our proposed framework in challenging neuroimaging tasks. The nature of these tasks poses several unique challenges, including relatively small datasets, multivariate non-stationary signals with complex dynamics, distribution shifts (severe across subjects; also indicated in Figure 4), low signal-to-noise ratios, and non-linear coupling mechanisms between EEG and fMRI. See Appendix L for more details.

### 4.1 SIMULATIONS

To study the effect of the $\varepsilon-$geodesic constraint and demonstrate that the proposed framework can recover the activity of a latent, shared source, we generated paired SPD matrix-valued data. That is, we generated a shared noise source with fixed, view-specific projections to $\mathcal{S}^2$ and paired it with a view-specific noise source. Parallel transport and exponential mapping were used to project the data to $\mathcal{S}^2_{++}$ with a random, view-specific Fréchet mean. Figure 3 summarizes the generative process and the considered architecture. For both modalities, we used SPDNet (Huang & Van Gool, 2017) as dimension adaptation and transformation module (Figure 3b). As tangent space projection module, we combine a SPD momentum batch norm (SPDMBN) layer (Kobler et al., 2022a) and LOG layer. For a thorough introductions of neural networks on SPD manifolds, please refer to Appendix G.

To quantify performance, we report conventional Euclidean vector space scores. For geometric approaches, we converted the latent SPD matrix-valued representations to Euclidean vector space features via tangent space projection. Conventional CCA was then applied to the training data to estimate one component. We report Pearson's correlation coefficients and coefficients of determination $R^2$ [2] on the held out test data.

Figure 3c summarizes cross validation (CV) results of our framework for various $\varepsilon$ compared to several baseline methods. The baseline methods were either fitted to the generated SPD data (CCA and RieCCA) or used as alternative loss terms (DeepCCA, VICReg) for the considered architecture. For $\varepsilon = 0$ (no geodesic constraint), DeepGeoCCA shared the highest correlation score with DeepCCA among the deep learning methods (left panel) at the cost of small explained variance (i.e., small $R^2$) in the latent representations (right panel). This result can be expected because both objectives only aim to maximize correlation in latent space. For $\varepsilon = 1$, the trained network traded a noticeable reduction in correlation for minimizing deviations from the geodesic constraint. Moreover, for some CV splits, the network got stuck in poor local minima (noticeable lower correlation and $R^2$). We found that $\varepsilon = 0.75$ resulted in a good trade-off between both objectives while avoiding poor local minima. Scatterplots in Figure 3b visualize the learned representations ($\varepsilon = 0.75$) for a representative CV split. Comparing the correlation scores of CCA and RieCCA, clearly demonstrate the importance of utilizing the mathematical structure of SPD matrices. Overall, RieCCA obtained the highest correlation score. This is likely because RieCCA had access to the entire training data to find the optimal solution for this toy problem, while the deep learning approaches were fitted to randomly sampled mini-batches.

---

[2] The coefficient of determination $R^2$ is used to quantify how much variance in the latent representations can be explained by the first CCA component, i.e., $R^2 = 1 \Leftrightarrow$ data distributed along a line or geodesic for geometric approaches.

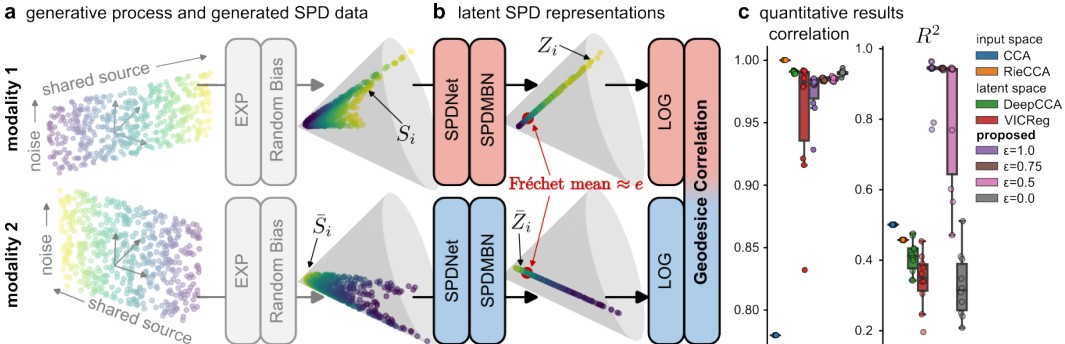

Figure 3: **Simulations on** $\mathcal{S}^2_{++}$. **a,** Generative process and visualization of generated observations ($N = 500$) with a shared latent source. **b,** Visualization of the latent representations learned with DeepGeoCCA ($\varepsilon = 0.75$). **c,** Boxplots summarize the simulation results for test data (10-fold CV). DeepGeoCCA can obtain high geodesic correlation (left panel; higher is better) and at the same time approx. constrain the representations to geodesics (right panel; coefficient of determination $R^2$; mean across modalities; higher is better).

## 4.2 SIMULTANEOUS EEG-FMRI DATA

The aim of this experiment is to exhibit the feasibility of DeepGeoCCA. Since EEG and fMRI provide distinct views to brain dynamics due to their diametral spatiotemporal sensitivities and underlying neurophysiological coupling mechanisms, the proposed approach is fundamentally different from previous data-driven approaches which used extensive domain-knowledge to extract low-dimensional representations (Deligianni et al., 2014; Dähne et al., 2015; Keynan et al., 2019; Wirsich et al., 2020; Abreu et al., 2021; Ogawa et al., 2021; Philiastides et al., 2021) or tried to transcode them (Liu et al., 2022). Using simultaneous EEG and fMRI measurements, we aim at learning latent representations where EEG and fMRI dynamics are highly correlated, and remain so for held out data. A publicly available simultaneous EEG-fMRI dataset (van der Meer et al., 2016a) is used to demonstrate feasibility. It contains data of 8 subjects while they were resting in two different conditions (eyes closed or open). Both conditions lead to changes in whole-brain dynamics, which are captured by EEG (Barry et al., 2007) and fMRI (Costumero et al., 2020), rendering the dataset an ideal candidate to evaluate our proposed framework against baseline methods. For a detailed description of the dataset, EEG and fMRI preprocessing and implementation details, please refer to Appendix K.

Figure 4 summarizes our experimental approach. We used t-SNE to visualize the preprocessed EEG and fMRI data $\{(S_i, \bar{S}_i)\}_{i=1}^N$ as well as the learned representations $\{(Z_i, \bar{Z}_i)\}_{i=1}^N$ for a representative CV split. For EEG data, we tested two architectures, both extracting latent representations in $\mathcal{S}^{10}_{++}$. The first one was similar to the architecture used in the simulations. The second one (TSM-Net) is conceptually similar except that it applies convolutional feature extractors to EEG time-series data before covariance pooling (Kobler et al., 2022a; Wilson et al., 2022). To accommodate large distribution shifts (e.g., in top-left t-SNE plot in Figure 4) across subjects and runs often encountered in EEG (Jayaram et al., 2016; Rodrigues et al., 2019; Wu et al., 2020; Défossez et al., 2023), we used subject and run-specific batch normalization layers (Kobler et al., 2022a). To facilitate a fair comparison with RieCCA and CCA, we re-centered the inputs for these methods per run according to (Zanini et al., 2018). For fMRI data, we used a standard transformer-encoder layer (Vaswani et al., 2017) which proved useful in learning latent representations from large-scale, diverse fMRI datasets (Thomas et al., 2022). This approach considered each fMRI volume inside a sliding window as a token. At the end of each window, we concatenated a learnable summary (SUM) token. Finally, a linear layer projected the transformed SUM token into a symmetric matrix which we used according to Remark 3 as $\log_e(\bar{Z}_i)$ in the overall loss (3).

Table 1 summarizes the results for a 10-fold cross-validation scenario. As in section 4.1, we use Pearson's correlation to assess the consistency between the learned EEG and fMRI representations and $R^2$ to quantify the geodesic constraints. Utilizing TSMNet in DeepGeoCCA obtained significantly higher correlation than all other candidates. Learning the considered architectures with alternative

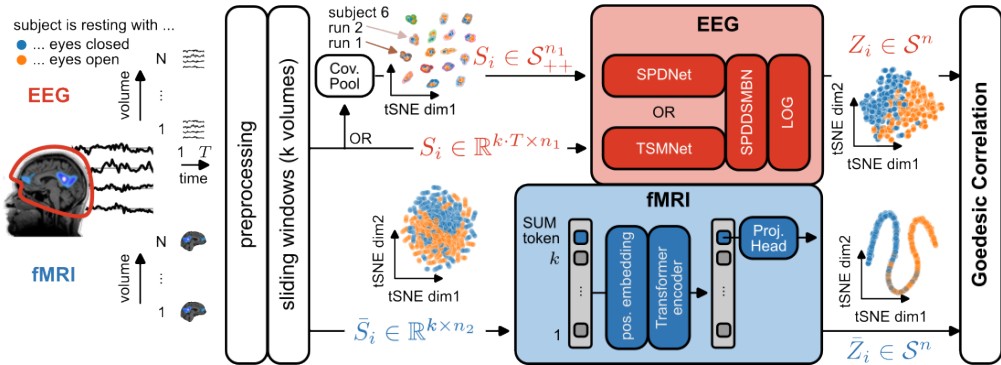

Figure 4: **Diagram of DeepGeoCCA for Simultaneous EEG-fMRI Data**. Using simultaneously recorded observations $(S_i, \bar{S}_i)$, we aim to learn a latent space where brain dynamics $(Z_i, \bar{Z}_i)$, shared between EEG and fMRI, covary with high-congruence. After preprocessing and extracting sliding windows, we use established decoder models to convert fMRI activity (Thomas et al., 2022) and oscillatory EEG activity (Kobler et al., 2022a) to latent representations. Visualizations (t-SNE, perplexity=30) summarize input and latent data distributions after fitting.

Table 1: **Results for Simultaneous EEG-fMRI Data**. Test-set model performance (higher is better) is evaluated with correlation and $R^2$ metrics across 10-fold CV with stratification across subjects and runs. Exhaustive permutation t-tests (df=9, 7 tests with t-max adjustment) were used to identify significant differences between TSMNet+DeepGeoCCA and baseline methods.

| fMRI model | EEG model | Method | Correlation ↑ | | $R^2$ ↑ | |
|---|---|---|---|---|---|---|
| | | | mean (std) | t-val (p-val) | mean (std) | t-val (p-val) |
| | | CCA | 0.02 (0.15) | -9.9 (0.002) | 0.07 (0.08) | -19.8 (0.002) |
| | | RieCCA | 0.38 (0.10) | -5.4 (0.002) | 0.01 (0.01) | -88.4 (0.002) |
| Transformer Encoder | SPDNet | DeepCCA | 0.42 (0.11) | -3.5 (0.031) | 0.30 (0.07) | -12.3 (0.002) |
| | | VICReg | 0.29 (0.19) | -4.0 (0.018) | 0.28 (0.10) | -9.2 (0.002) |
| | | DeepGeoCCA | 0.44 (0.11) | -3.0 (0.049) | **0.58** (0.02) | 3.8 (0.023) |
| | TSMNet | DeepCCA | 0.29 (0.10) | -8.8 (0.002) | 0.18 (0.09) | -15.2 (0.002) |
| | | VICReg | 0.38 (0.10) | -5.5 (0.002) | 0.24 (0.03) | -40.5 (0.002) |
| | | DeepGeoCCA | **0.58** (0.08) | / | 0.55 (0.02) | / |

loss terms (VICReg, DeepCCA) resulted in obvious performance drops for either or both EEG architectures. Moreover, compared to the performance of RieCCA, the results clearly demonstrate the utility of DeepGeoCCA to extract shared, latent brain dynamics whose coupling generalizes to unseen data. Note that cross subject generalization results, listed in Table 7 of Appendix K, show similar effects, confirming the robustness of DeepGeoCCA. Taking a look at the EEG architectures, the results suggest an advantage of combining learnable convolutional layers with latent covariance pooling over broad-band EEG spatial covariance matrices as inputs.

## 4.3 MULTI-VIEW EEG DATA

The aim of this experiment is to evaluate the performance of DeepGeoCCA in a downstream task, namely EEG-based motor imagery classification. During planning, executing but also imagining limb movements, EEG captures spatio-spectrally localized fluctuations in sensorimotor rhythms (Pfurtscheller & Da Silva, 1999). These fluctuations give rise to identifiable patterns that consistently enable classification of EEGs.

Motivated by the emergence of mobile, few-channel EEG devices, we generate pairs of two-channel EEG and multi-channel (full sensorimotor coverage) EEG views. We use DeepGeoCCA for model pre-training with the goal to improve downstream classification performance of the 2-channel EEG views. Here, we kept geodesic speeds $v = w$ fixed and initialized them with tangent space principal component directions of the EEG spatial covariance matrices of the full-channel EEG views. For detailed settings of this experiment, please refer to Appendix J.

Using DeepGeoCCA to pre-train Tensor-CSPNet (Ju & Guan, 2022) resulted in statistically significant group-level improvements in both considered datasets (Table 2 and Supplementary Table 5), compared to the lower bound (i.e., two EEG channels without SSL pre-training). Although our proposed loss consistently yielded significant improvements, the effect remained small across scenarios compared to other SSL pre-training losses. Nonetheless, the results clearly indicate that our framework for pre-training SPD neural nets like Tensor-CSPNet with paired views can yield performance increases in downstream tasks. For detailed results at the subject-level, refer to Figures 9 and 10 in Appendix J.

Table 2: **Downstream Results for Multi-view EEG Data**. Average (std across subjects in brackets) classification accuracy for the motor imagery task (higher is better). We considered two public datasets, and two scenarios, namely, within-session 10-fold CV (sessions $S_1$ and $S_2$) and a hold-out scenario (train $S_1$; test $S_2$). For the KU and BNCI2015001 datasets the all channel EEG views comprise 20 and 13 channels above the sensorimotor cortex. Tensor-CSPNet is utilized as the network architecture in DeepGeoCCA.

| Channel | SSL pre-training | KU (54 subjects, 2 classes) | | | BNCI2015001 (12 subjects, 2 classes) | | |
|---|---|---|---|---|---|---|---|
| | | $S_1$ | $S_2$ | $S_1 \rightarrow S_2$ | $S_1$ | $S_2$ | $S_1 \rightarrow S_2$ |
| Two | ✗ | 62.97 (15.72) | 65.85 (16.03) | 62.89 (12.18) | 74.12 (18.20) | 75.62 (14.22) | 75.00 (14.10) |
| | DeepCCA | 66.24 (15.73) | 68.11 (15.67) | 65.20 (13.24) | 76.46 (16.83) | 78.33 (11.33) | 76.58 (13.23) |
| | VICReg | 66.12 (15.80) | 67.35 (16.80) | 65.20 (13.51) | 77.21 (15.96) | 77.25 (13.46) | 77.00 (13.97) |
| | DeepGeoCCA | **66.26** (17.05) | **68.27** (17.23) | **65.93** (13.68) | **77.58** (16.26) | **78.46** (12.82) | **77.83** (13.11) |
| All | ✗ | 70.56 (16.26) | 72.86 (15.86) | 68.85 (14.18) | 84.33 (10.10) | 86.46 (14.19) | 85.17 (11.31) |

## 5 DISCUSSION

Motivated by wide-spread applications of covariance-based neuroimaging data, we capture the correlation consistency of paired views on SPD manifolds using a novel measure called geodesic correlation. This novel correlation is a generalized concept of correlation for SPD matrix-valued data. It is well suited for deep neural networks on SPD manifolds compared to conventional correlation measures. To maximize geodesic correlation in a latent space between paired views of SPD matrix-valued representations, we propose a novel loss function using the relaxed geodesic constraints, and present a novel geometric deep learning-based SSL framework, referred to as DeepGeoCCA. DeepGeoCCA is more generally applicable than classical non-deep learning approaches, because, in a general sense, deep learning is capable of handling datasets with intricate high-dimensional data distributions, and align latent representations of neural data collected from a diverse population. In particular, our framework respects the geometric structure of SPD matrix-valued data. In simulations and experiments with EEG and fMRI measurements, DeepGeoCCA can learn representations whose dynamics generalize favorably to held-out data, while preserving task-relevant information.

Overall, this work can be regarded as a proof-of-concept that geometric deep learning-based SSL can be used to extract meaningful representations from simultaneous EEG-fMRI measurements solely from paired observations. Although experiments with EEG-fMRI data is limited to a small dataset with task-related effects in either modality, our work is the first completely data-driven SSL framework in this context. In practice, one may expect further gains by utilizing large-scale datasets and uni-modal SSL pre-training objectives proposed in earlier studies, for example Thomas et al. (2022); Banville et al. (2020).

Besides, for applications beyond the considered tasks, it is necessary to identify interesting factors of variation to generate paired views. In a clinical setting, a task could be disease diagnosis. Naturally, views associated with the same subject could be considered as paired. For example, our framework could contribute to relating functional, derived from fMRI, and anatomical, derived from diffusion tensor imaging, connectomes with the goal to identify latent, potentially clinically relevant, modes of variation across subjects.

## ACKNOWLEDGMENTS

This work was supported by Innovative Science and Technology Initiative for Security Grant Number JPJ004596, ATLA and KAKENHI (Grants-in-Aid for Scientific Research) under Grant Numbers 21H03516, 21K12055, JSPS, Japan.

This work was supported under the RIE2020 Industry Alignment Fund–Industry Collaboration Projects (IAF-ICP) Funding Initiative, as well as cash and in-kind contributions from industry partner(s); This work was also supported by the RIE2020 AME Programmatic Fund, Singapore (No. A20G8b0102).

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

## A  SPD MANIFOLDS WITH AFFINE INVARIANT RIEMANNIAN METRIC $g^{AIRM}$

In the section, we will elucidate covariance-based neuroimaging data on the SPD manifold. First, we present its definition as follows,

**Definition** (Covariance-Based Neuroimaging Data). *The covariance matrix of a neuroimaging data segment, represented by $X \in \mathbb{R}^{n_C \times n_T}$, is denoted as $S = XX^\top / n_T \in \mathcal{S}_{++}^{n_C}$, where $n_C$ is the number of spatial dimensions and $n_T$ is the number of timesteps.*

Covariance-based data is inherently symmetric and positive definite. Even for degenerate cases (e.g., co-linear spatial dimension), the positive definiteness can be ensured via shrinkage regularization Honnorat & Habes (2022).

Note that this definition holds for any multi-channel neuroimaging time series. However, in this study, we focus on two neural signal modalities, namely EEG spatial covariance matrices and fMRI functional connectivities. EEG spatial covariance matrices capture relevant aspects of neural activity within short to long-time scales. Riemannian geometry-aware methods operating with these covariance-based data are considered state-of-the-art in various EEG application domains, including clinical applications (Sabbagh et al., 2020; Gemein et al., 2020) and recent brain-computer interface competitions (Roy et al., 2022). Concerning fMRI, covariance (or correlation) matrices between regions of interest reflect static and dynamic communication between brain networks and their alterations in tasks (Park & Friston, 2013) and pathological conditions (Yamashita et al., 2020).

Equipped with the affine invariant Riemannian metric $g^{AIRM}$, as follows,

$$g_P^{AIRM}(v, w) := \langle P^{-\frac{1}{2}} v P^{-\frac{1}{2}}, P^{-\frac{1}{2}} w P^{-\frac{1}{2}} \rangle_{\mathcal{F}} = \mathrm{Tr}(P^{-1} v P^{-1} w),$$

for any tangent vectors $v$ and $w$ in the tangent space $\mathcal{T}_P \mathcal{S}_{++}$ at the base point $P \in \mathcal{S}_{++}$, the space of covariance-based neuroimaging data becomes an SPD manifold, denoted as $(\mathcal{S}_{++}, g^{AIRM})$, which is a Cartan-Hadamard manifold. A Cartan-Hadamard manifold is a Riemannian manifold that is complete, connected, and possesses an everywhere non-positive sectional curvature. In particular, the Riemannian distance between two matrices $S_1$ and $S_2$ is defined as follows,

$$d_{g^{AIRM}}(S_1, S_2) = \left\| \log_e(S_1^{-1} \cdot S_2) \right\|_{\mathcal{F}}. \tag{4}$$

## B  FRÉCHET MEAN AND VARIANCE

In this section, we provide a brief introduction to the Fréchet mean and variance (Fréchet, 1948), also known as Karcher mean (Grove & Karcher, 1973). The two names are often used interchangeably in literature.

Given a set of points $\{p_1, ..., p_N\} \in (\mathcal{M}, g)$, the Fréchet mean $\mu$ is defined as the minimizer of the sum-of-squared Riemannian distances as follows,

$$\mu := \arg \min_{p \in \mathcal{M}} \frac{1}{N} \sum_{i=1}^{N} d_g^2(p, p_i). \tag{5}$$

And, the corresponding Fréchet variance is defined as follows,

$$\sigma^2 := \mathcal{E}\left[ d_g^2(\mu, p) \right] = \frac{1}{N} \sum_{i=1}^{N} d_g^2(\mu, p_i). \tag{6}$$

**Remark.** *We consistently employ parallel transport to relocate the Riemannian means of each data cluster to the identity matrix $e$. Consequently, the Fréchet variance can be simplified as follows:*

$$\sigma^2 = \frac{1}{N} \sum_{i=1}^{N} \left\| \log_e(S_i) \right\|_e^2,$$

*which is given by $d_{g^{AIRM}}(e, S) = \left\| \log_e(S) \right\|_e$, where Riemannian norm $\|v\|_e := \sqrt{g_e^{AIRM}(v, v)}$ for any $v \in \mathcal{T}_e \mathcal{S}_{++}$.*

## C  ORTHOGONAL PROJECTION AND GEODESIC CONSTRAINTS

This section first provides an elementary introduction to the geodesic on Riemannian manifolds.

Let $(\mathcal{M}^n, g)$ be a Riemannian manifold with dimension $n$ equipped with the Levi-Civita connection $\nabla$, and $\gamma : \mathbb{R} \longmapsto \mathcal{M}$ be a smooth curve on $\mathcal{M}$. We say a vector field $X$ is parallel along $\gamma$ if $\nabla_{\gamma'(t)} X = 0$ for all $t \in \mathbb{R}$. A map $P_{t_0,t}^{\gamma} : \mathcal{T}_{\gamma(t_0)}\mathcal{M} \longmapsto \mathcal{T}_{\gamma(t)}\mathcal{M}$ is a parallel transport of $X_0 \in \mathcal{T}_{\gamma(t_0)}\mathcal{M}$ along $\gamma$ if $P_{t_0,t}^{\gamma}(X(\gamma(t)))$ is the parallel vector field along $\gamma(t)$ such that $X(\gamma(0)) := X_0$.

We say $\gamma(t)$ is a *geodesic* if $\gamma'(t)$ is parallel along $\gamma(t)$, i.e.,

$$\nabla_{\gamma'(t)} \gamma'(t) = 0.$$

Note that $\nabla_{\gamma'(t)} \gamma'(t) = 0$ enforces $\gamma(t)$ to be curved with zero acceleration. The concept *geodesic* extends the idea of straight lines from Euclidean spaces. It is worth mentioning that every geodesic is parameterized in proportion to its arc length.

In a local chart $(\mathcal{U}, \varphi)$, the geodesic curve $\gamma(t)$ satisfies the following second-order ordinary differential equations:

$$\ddot{u}_k(t) + \sum_{ij} \Gamma_{ij}^k \dot{u}_i(t) \dot{u}_j(t) = 0, \ \text{ for } k = 1, ..., n,$$

where each $u_k(t) = \varphi^k \circ \gamma(t)$. For local frames $\{\partial_i\}_{i=1}^n$, $\Gamma_{ij}^k$ denotes the Christoffel symbols of $\nabla$ given by $\nabla_{\partial_i}(\partial_j) = \Gamma_{ij}^k \partial_k$, $\forall\, i, j, k = 1, ..., n$. By the Picard-Lindelöf theorem, it yields the local existence and uniqueness of the solution for a geodesic: For any given $p \in \mathcal{M}$ and $v \in \mathcal{T}_p\mathcal{M}$, there exists an interval $(-\epsilon, \epsilon)$ and a unique geodesic $\gamma : (-\epsilon, \epsilon) \longmapsto \mathcal{M}$ satisfying the initial conditions $\gamma(0) := p$ and $\gamma'(0) := v$.

Given that the SPD manifold $(\mathcal{S}_{++}, g^{AIRM})$ is geodesically complete, meaning that all geodesics are defined for all time $t \in \mathbb{R}$, any geodesic $\exp_e(tu)$ on $\mathcal{S}_{++}$ is solely determined by $u \in \mathcal{T}_e\mathcal{S}_{++}$ and varies smoothly with respect to $u$.

To clarify the geodesic constraints, we will begin by introducing the idea of orthogonal projection: given an unit speed $u \in \mathcal{S}^n$, the orthogonal projection of $S \in \mathcal{S}_{++}^n$ on arc-length geodesic $\exp_e(tu) : \mathbb{R} \longmapsto (\mathcal{S}_{++}^n, g^{AIRM})$ is $S^\dagger = \exp_e(t^\dagger u)$ [3] with the following optimality condition:

$$t^\dagger := \arg \min_{t \in \mathbb{R}} d_{g^{AIRM}}^2 \big(S, \exp_e(tu)\big). \tag{7}$$

Plugging into Equation 4, Equation (7) becomes into the following expression:

$$t^\dagger := \arg \min_{t \in (-\epsilon, \epsilon)} \mathrm{Tr}\Big( \log_e^2 \big(S^{-1} \cdot \exp_e(tu)\big)\Big). \tag{8}$$

Take the derivative of Equation (8) with respect to time $t$, set it to zero, and we obtain:

$$0 = \frac{\partial}{\partial t} \mathrm{Tr}\Big( \log_e^2 \big(S^{-1} \cdot \exp_e(tu)\big)\Big)$$
$$\overset{(*)}{=} 2 \cdot \mathrm{Tr}\Big( \log_e \big(S^{-1} \cdot \exp_e(tu)\big) \cdot \exp_e(-tu) \cdot \frac{\partial}{\partial t} \exp_e(tu)\Big),$$

which yields an equality as follows,

$$\mathrm{Tr}\Big( \log_e \big(S^{-1} \cdot \exp_e(tu)\big) \cdot \exp_e(-tu) \cdot u \cdot \exp_e(tu)\Big) = 0. \tag{9}$$

Equality $(*)$ is valid by virtue of the following proposition:

**Proposition 1** (Moakher (2005)). *Consider a real matrix-valued function $S(t)$ defined on time $t \in \mathbb{R}$. Assume that $S(t)$ is an invertible matrix and does not have eigenvalues on the closed negative real line. Then, we have the following equality:*

$$\frac{\partial}{\partial t} \mathrm{Tr}\Big( \log_e^2 S(t)\Big) = 2 \cdot \mathrm{Tr}\Big( \log_e S(t) \cdot S^{-1}(t) \cdot \frac{\partial}{\partial t} S(t)\Big).$$

[3] A covariance matrix $S$ marked with a $\dagger$ symbol in the upper right corner denoted as $S^\dagger$, signifies that it represents the orthogonal projection of $S$ on the geodesic.

Suppose $t := t^\dagger$ is optimal and thus $S^\dagger := \exp_e(t^\dagger u)$. Hence, Equality (9) becomes to be as follows,

$$\text{Tr}\Big( \log_e \big( S^{-1} \cdot S^\dagger \big) \cdot S^{\dagger^{-1}} \cdot u \cdot S^\dagger \Big) = 0. \tag{10}$$

According to the problem setting given in Definition 3, Equation 10 yields the following two geodesic constraints of paired covariance-based neuroimaging data for all $i \in \{1, ..., N\}$:

$$\text{Tr}\Big( \log_e \big( S_i^{-1} \cdot S_i^\dagger \big) \cdot S_i^{\dagger^{-1}} \cdot v \cdot S_i^\dagger \Big) = 0;$$

$$\text{Tr}\Big( \log_e \big( \bar{S}_i^{-1} \cdot \bar{S}_i^\dagger \big) \cdot \bar{S}_i^{\dagger^{-1}} \cdot w \cdot \bar{S}_i^\dagger \Big) = 0.$$

**Remark.** *In this remark, we briefly discuss the reason for introducing geodesic and the orthogonal projection on the geodesic. The concept of geodesic constraint originates from the field of Geometric Statistics, referred to as the geodesic submanifolds, initially introduced in principal geodesic analysis (PGA) and related works (Fletcher et al., 2004; Fletcher & Joshi, 2007; Thomas Fletcher, 2013).*

*In the context of Riemannian geometry, a geodesic is a curve that locally represents the shortest path between points, acting as a generalization of a straight line. Moreover, the geodesic subspace can be viewed as an expansion of linear spaces from Euclidean to Riemannian geometry. The concept of the geodesic subspace is pivotal in PGA as the projection on this subspace preserves the Riemannian distance, representing the manifold-valued data's variance.*

*In a formal sense, a submanifold $H$ of manifold $M$ is considered geodesic at the point $p \in H$ if all geodesics of $N$ that pass through $p$ remain geodesic of $M$. A geodesic submanifold $H$ at the Fréchet mean $\mu$ is represented as the exponential mapping of the linear span of $r$ tangent vectors $\{w_i\}_{i=1}^r \in \mathcal{T}_\mu M$ as $N = \exp_\mu \big( span(\{w_i\}_{i=1}^r) \big)$.*

*In PGA, tangent vectors $\{w_i\}_{i=1}^r$ are computed by sequentially maximizing the Riemannian distance between each data point and the Fréchet mean. This process captures the variance of the data at each step while progressively removing the influence of previously derived directional projections. This completes the manifold-value data's dimensionality reduction. In this context, $w_1$ can be analogous to the first principal component in principal component analysis (PCA). This work goes beyond finding the geodesic submanifold's tangent vectors $\{w_i\}_{i=1}^r$, obtaining a novel low-dimensional data representation is crucial.*

*CCA aims to identify the maximum correlation between two variables. Specifically, as outlined in Section 2.2, CCA aims to find linear transformations for paired matrices $X \in \mathbb{R}^{N \times p}$ and $\bar{X} \in \mathbb{R}^{N \times q}$, denoted as $w \in \mathbb{R}^p$ and $\bar{w} \in \mathbb{R}^q$, respectively, to maximize the correlation between these two linear combinations $Xw$ and $\bar{X}\bar{w}$. CCA is typically transformed into an eigenvalue decomposition problem using the Lagrange multipliers method. Consequently, the objective often involves identifying a set of $r$ pairs of canonical variables along with corresponding linear transformations $W \in \mathbb{R}^{p \times r}$ and $\bar{W} \in \mathbb{R}^{q \times r}$, where each column in these matrices signifies a mapping to the respective pair of the canonical covariate. A constraint is imposed to render them uncorrelated to one another to ensure that each pair of canonical variables captures distinct phenomena.*

*When extending the CCA method to Riemannian manifolds, Kim et al. (2014) was the first to use geodesic submanifolds as a replacement for linear subspaces. They did not underscore the orthogonal constraints for each pair of canonical covariates and obtained them through a nonlinear optimization approach. Each resulting pair of canonical covariates corresponds to tangent vectors $\{w_i\}_{i=1}^r$ and $\{\bar{w}_i\}_{i=1}^r$ for two geodesic submanifolds. Therefore, orthogonal projection on the geodesic can be seen as projecting the data onto a lower-dimensional space by analogy with a linear transformation in classical CCA. Each tangent vector in the tangent space corresponds to a column vector in the linear transformation (canonical covariates).*

*The initial target of this study was to enhance the temporal dynamic consistency of simultaneously recorded EEG-fMRI signals. We opt for only one geodesic path for each modality to streamline the method. This design simplification indeed leads to a well-defined convex problem and computational advantages in our deep learning architecture. In particular, opting for only one geodesic path for each modality is equivalent to utilizing the first pair of canonical covariates in CCA. In other words, the tangent vector $w_1$, a symmetric matrix (tangent vector for SPD manifolds), precisely serves as the first canonical covariate. Thus, while the mechanism for obtaining $w_1$ differs from that obtained*

*through eigenvalue decomposition in CCA or DeepCCA, our proposed component analysis method inherently possesses a directional aspect.*

## D COMPARISONS BETWEEN DIFFERENT CCAS

In Table 3, we analyze the similarities and differences between several related variants of the CCA method and our proposed method from three perspectives, including:

- Nonlinear Projection: Projections on the nonlinear subspace or not.
- Eigendecomposition: Utilizing the eigendecomposition method or not, i.e., eigendecomposition of $\left\|\sum_{xx}^{-1/2} \cdot \sum_{x\bar{x}} \cdot \sum_{\bar{x}\bar{x}}^{-1/2}\right\|_{\mathcal{F}}$ or $\left\|\sum_{\varphi\varphi}^{-1/2} \cdot \sum_{\varphi\bar{\varphi}} \cdot \sum_{\bar{\varphi}\bar{\varphi}}^{-1/2}\right\|_{\mathcal{F}}$ in Subsection 2.2 and 2.3.
- Neural Networks-Based Approach: the neural networks-based approach or not.
- Geometry: SPD matrix-valued outputs or not.

Table 3: Comparisons between different CCAs.

| Methodology | Nonlinear Projection | Eigendecomposition | Neural Networks | Geometry |
|---|---|---|---|---|
| CCA | × | ✓ | × | × |
| DeepCCA | ✓ | ✓ | ✓ | × |
| RieCCA | ✓ | ✓ | × | ✓ |
| DeepGeoCCA | ✓ | × | ✓ | ✓ |

Please note that RieCCA is solved analytically through an approximation approach within the log-Euclidean framework using CCA (Kim et al., 2014). Therefore, we say that RieCCA involves eigendecomposition.

## E CORRELATION, RIEMANNIAN CORRELATION, AND GEODESIC CORRELATION

In this chapter, we will provide a detailed explanation of the distinctions between correlation, Riemannian correlation, and geodesic correlation. Figure 5 illustrates the distinctions between the three terms. All of them are utilized to elucidate statistical relationships between two random variables, whether causal or not. However, their definitions encompass different scopes, as outlined below:

- Correlation: it is a general statistical term for any random variable (Croxton, 1967).
- Riemannian Correlation: it is a generalized term of correlation for general Riemannian manifolds. It is quantified by computing the correlation between tangent vectors, which represent manifold-valued data projected onto the respective geodesic submanifold of the tangent spaces at the Fréchet mean of each modality (Kim et al., 2014).
- Geodesic Correlation: it is a generalized term of correlation for covariance-based neuroimaging data. It is quantified by calculating the correlation between tangent vectors, which represent manifold-valued data projected onto the respective geodesics of the tangent spaces of each modality at the identity matrix $e$.

The main motivation for introducing geodesic correlation is due to the fact that $(\mathcal{S}_{++}, g^{AIRM})$ is a Cartan-Hadamard manifold that is complete, connected, and has an everywhere non-positive sectional curvature (Pennec, 2020). Consequently, $(\mathcal{S}_{++}, g^{AIRM})$ is geodesically complete, and the geodesics can be extended to the whole $\mathbb{R}$. These excellent geometric properties enable the simplification of Riemannian correlation on the SPD cone. We utilize this fact as follows:

- Geodesics: We simplify the constraint of a geodesic submanifold in Riemannian correlation to a geodesic.

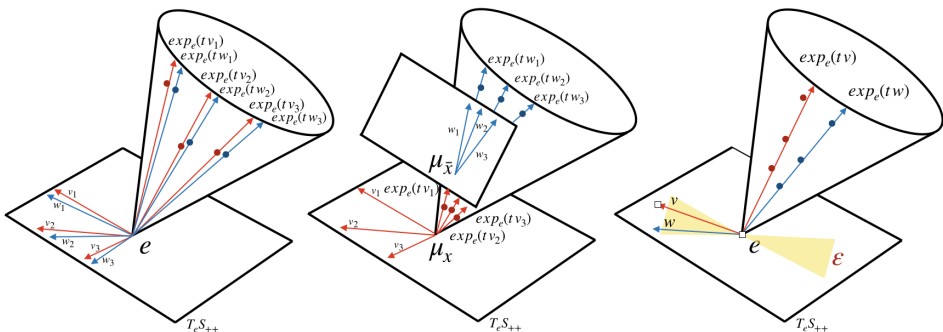

Figure 5: Illustrations of Correlation, Riemannian Correlation, and Geodesic Correlation: From left to right, we have Correlation, Riemannian Correlation, and Geodesic Correlation with the specific descriptions. **Correlation**: Correlation is only required to be demonstrated between paired data; Between each pair of paired data, there is no need for proximity; **Riemannian Correlation**: It computes the correlation between paired projection data on the tangent spaces of each modality, considering their respective Fréchet means. The projection data is obtained by projecting the raw data onto geodesic submanifolds; **Geodesic Correlation**: It calculates the correlation between paired projection data in the tangent space at the identity point $e$. The projection data is obtained by projecting the raw data onto or near geodesics.

- Base Point $e$ of Tangent Space: We exclusively compute the correlation on the tangent space at the identity point $e$, rather than the Fréchet mean for each modality. In particular, defining geodesic correlation at $e$ is equivalent to defining it at any point $s \in \mathcal{S}_{++}$. This equivalence is established through parallel transportation (see Appendix G.2).
- Correlation Measurement: We use Riemannian metric to measure the correlation between two tangent vectors on the tangent space at the identity point $e$. In contrast, Riemannian Correlation relies on the inner product of scalar values of tangent vectors based on different Fréchet means.

Given these simplifications, geodesic correlation fulfills the requirements of neural network-based solution. We summarize the primary distinctions from the four aspects discussed earlier in Table 4 as follows:

Table 4: Difference between Correlation, Riemannian Correlation and Geodesic Correlation

| Correlation | Region | Base Point | Correlation Measurement | Solver |
|---|---|---|---|---|
| Correlation | Lines | 0 | Dot product | CCA |
| Riemannian Correlation | Geodesic submanifolds | Two Fréchet means | Scalar dot product | RieCCA |
| Geodesic Correlation | Geodesics | $e$ | Riemannian metric | DeepGeoCCA |

Geodesic correlation has a relationship with correlation according to the following proposition:

**Proposition 2.** *Suppose $v$ and $w$ are tangent vectors on $\mathcal{T}_e\mathcal{S}_{++}^n$ of $(\mathcal{S}_{++}^n, g^{AIRM})$, then we have*

$$g_e^{AIRM}(v,w) = \langle vec(v), vec(w) \rangle_{\ell^2}.$$

*Proof.* This proposition arises from the following derivation,

$$g_e^{AIRM}(v,w) := \text{Tr}(v \cdot w) = \sum_{i=1}^{N} v[i,:] \cdot w[:,i] = \langle vec(v^\top), vec(w) \rangle_{\ell^2} \stackrel{(*)}{=} \langle vec(v), vec(w) \rangle_{\ell^2},$$

where $(*)$ is derived from the property that the tangent space of $(\mathcal{S}_{++}^n, g^{AIRM})$ is a symmetric space. □

This proposition demonstrates that maximizing the geodesic correlation on $(\mathcal{S}_{++}^n, g^{AIRM})$ at the base point $e$ is equivalent to maximizing the correlation if we view the covariance-based data as

a flattened tensor. Furthermore, this proposition also implies that the coefficient of determination $R^2$ in the Riemannian geometry-based sense, as discussed in Fletcher (2011), is equivalent to the classical $R^2$.

## F  CONSTRAINED NONLINEAR OPTIMIZATION PROBLEM

This section will show how to convert geodesic correlation into a constrained nonlinear optimization problem in matrix and Euclidean space.

In matrix space, the constrained nonlinear optimization problem is depicted as follows,

$$
\begin{aligned}
\underset{v, w, S_i^\dagger, \bar{S}_i^\dagger}{\text{minimize}} \quad & -\frac{\sum_{i=1}^N g_e^{AIRM}\big(\log_e(S_i^\dagger), \log_e(\bar{S}_i^\dagger)\big)}{\left(\sum_{i=1}^N \left\|\log_e(S_i^\dagger)\right\|_e^2\right)^{\frac{1}{2}} \cdot \left(\sum_{i=1}^N \left\|\log_e(\bar{S}_i^\dagger)\right\|_e^2\right)^{\frac{1}{2}}} \\
\text{subject to} \quad & \mathrm{Tr}\Big( \log_e\big(S_i^{-1}\cdot S_i^\dagger\big) \cdot S_i^{\dagger^{-1}} \cdot v \cdot S_i^\dagger \Big) = 0 \\
& \mathrm{Tr}\Big( \log_e\big(\bar{S}_i^{-1}\cdot \bar{S}_i^\dagger\big) \cdot \bar{S}_i^{\dagger^{-1}} \cdot w \cdot \bar{S}_i^\dagger \Big) = 0
\end{aligned}
\tag{11}
$$

with variables $v, w \in \mathcal{S}^n$ and $S_i^\dagger, \bar{S}_i^\dagger \in \mathcal{S}_{++}$ for $i = 1, ..., N$. The consistency targeted by this constrained nonlinear optimization problem using correlation on the tangent space is referred to as geodesic correlation.

The constrained nonlinear optimization problem can also be written in Euclidean space. However, when the results obtained through a constrained nonlinear optimization algorithm in Euclidean space are converted back into matrices, they cannot be guaranteed to be positive definite and symmetric.

Next, we will derive the constrained nonlinear optimization problem in Euclidean space, equivalent to Problem (11) in the matrix space.

We substitute the correlation on $(\mathcal{S}_{++}^n, g^{AIRM})$ with conventional correlation denoted as follows,
$$
t_v^i := vec\big(\log_e(S_i)\big), \text{ and } t_w^i := vec\big(\log_e(\bar{S}_i)\big), \text{ for } i = 1, ..., N.
$$
Then, we obtain the constrained nonlinear optimization problem in Euclidean space with a total of $2N$ constraints, depicted as follows,

$$
\begin{aligned}
\underset{v, w, t_v^i, t_w^i}{\text{minimize}} \quad & -\frac{\sum_{i=1}^N \langle t_v^i, t_w^i \rangle}{\left(\sum_{i=1}^N \|t_v^i\|_{\ell^2}^2\right)^{\frac{1}{2}} \cdot \left(\sum_{i=1}^N \|t_w^i\|_{\ell^2}^2\right)^{\frac{1}{2}}} \\
\text{subject to} \quad & \mathrm{Tr}\Big( \log_e\big(S_i^{-1}\cdot \exp_e(t_v^i v)\big) \cdot \exp_e(-t_v^i v) \cdot v \cdot \exp_e(t_v^i v) \Big) = 0 \\
& \mathrm{Tr}\Big( \log_e\big(\bar{S}_i^{-1}\cdot \exp_e(t_w^i w)\big) \cdot \exp_e(-t_w^i w) \cdot w \cdot \exp_e(t_w^i w) \Big) = 0
\end{aligned}
\tag{12}
$$

with variables $v, w \in \mathcal{S}^n$ and $t_v^i, t_w^i \in \mathbb{R}^{n^2}$ for $i = 1, ..., N$.

By solving Constrained Nonlinear Optimization Problem (12) through constrained nonlinear optimization algorithms, even when the resulting $t_v$ and $t_w$ are converted into a matrix, it no longer maintains the positive definite symmetric properties.

In the following, we will derive the Lagrangian for Constrained Nonlinear Optimization Problem (11) in the matrix space. We arrive at the expression for Lagrangian $\mathcal{L}(S^\dagger, \bar{S}^\dagger, v, w, \lambda, \gamma)$ : $\mathcal{S}_{++}^n \times \mathcal{S}_{++}^n \times \mathcal{S}^n \times \mathcal{S}^n \times \mathbb{R}^N \times \mathbb{R}^N \longmapsto \mathbb{R}$, as follow,

$$
\begin{aligned}
\mathcal{L}(S^\dagger, \bar{S}^\dagger, v, w, \lambda, \gamma) := & -\frac{\sum_{i=1}^N g_e^{AIRM}\big(\log_e(S_i^\dagger), \log_e(\bar{S}_i^\dagger)\big)}{\left(\sum_{i=1}^N \left\|\log_e(S_i^\dagger)\right\|_e^2\right)^{\frac{1}{2}} \cdot \left(\sum_{i=1}^N \left\|\log_e(\bar{S}_i^\dagger)\right\|_e^2\right)^{\frac{1}{2}}} \\
& + \sum_{i=1}^N \lambda_i \cdot \mathrm{Tr}\Big( \log_e\big(S_i^{-1}\cdot S_i^\dagger\big) \cdot S_i^{\dagger^{-1}} \cdot v \cdot S_i^\dagger \Big) \\
& + \sum_{i=1}^N \gamma_i \cdot \mathrm{Tr}\Big( \log_e\big(\bar{S}_i^{-1}\cdot \bar{S}_i^\dagger\big) \cdot \bar{S}_i^{\dagger^{-1}} \cdot w \cdot \bar{S}_i^\dagger \Big),
\end{aligned}
$$

where $\{\lambda_i \in \mathbb{R}\}_{i=1}^N$ and $\{\gamma_i \in \mathbb{R}\}_{i=1}^N$ are Lagrange multipliers.

In the context that follows, we refer to the orthogonal projection $S^\dagger$ or $\bar{S}^\dagger$ as $Z$ or $\bar{Z}$, respectively. [4] These orthogonal projections are obtained by neural network-based training using gradient descent. In practical implementation, these values precede the input to the LOG layer. The results produced by the LOG layer are denoted as $\log_e(Z_i)$ and $\log_e(\bar{Z}_i)$.

Hence, the Lagrangian duality is naturally a loss function for DeepGeoCCA, depicted as follows,

$$\mathcal{L}(Z, \bar{Z}) := -\frac{\sum_{i=1}^N \mathrm{Tr}\big(\log_e(Z_i) \cdot \log_e(\bar{Z}_i)\big)}{\sqrt{\sum_{i=1}^N \mathrm{Tr}\big(\log_e^2(Z_i)\big)} \cdot \sqrt{\sum_{i=1}^N \mathrm{Tr}\big(\log_e^2(\bar{Z}_i)\big)}}$$
$$+ \sum_{i=1}^N \lambda_i \cdot \mathrm{Tr}^2\Big(\log_e\big(S_i^{-1} \cdot Z_i\big) \cdot Z_i^{-1} \cdot v \cdot Z_i\Big)$$
$$+ \sum_{i=1}^N \gamma_i \cdot \mathrm{Tr}^2\Big(\log_e\big(\bar{S}_i^{-1} \cdot \bar{Z}_i\big) \cdot \bar{Z}_i^{-1} \cdot w \cdot \bar{Z}_i\Big),$$

where $\lambda_i, \gamma_i \in \mathbb{R}$ are preset coefficients, for $i = 1, \ldots, N$, and $v, w \in \mathcal{S}^n$ are preset tangent vectors.

## G  NEURAL NETWORKS ON SPD MANIFOLDS

The SPD matrix-valued network layers within the self-supervised learning framework in the proposed architecture are outlined in the following three studies:

### G.1  SPDNET

Huang & Van Gool (2017) introduced the SPDNet network architectures which is specifically designed to operate with SPD matrices to ensure that the essential properties of symmetry and positive definiteness are maintained throughout the learning process. It incorporates the following layers:

- BiMap: This layer performs the bi-map transformation $W \cdot S \cdot W^\top$ on $S$. The transformation matrix $W$ is typically required to have a full-row rank.
- ReEig: This layer performs $U \cdot \max(\epsilon \cdot e, \Sigma) \cdot U^\top$.
- LOG: This layer maps $S$ to its tangent space at the identity matrix $e$ using $U \cdot \log_e(\Sigma) \cdot U^\top$,

where signal covariance matrix $S = U \cdot \Sigma \cdot U^\top$.

### G.2  BATCH NORMALIZATION

Brooks et al. (2019) conceived a layer, termed Riemannian BN, as a form of SPD matrix-valued network architecture to center and re-bias batches of latent SPD data $\mathcal{B}$. Formally, the computation of the batch Fréchet mean, denoted as $B$, involves solving (5) and utilizing parallel transport (Yair et al., 2019) to establish connections between each element $Z \in \mathcal{B}$ and the identity matrix $e$. This is achieved through the following operations:

$$\Gamma_{B \mapsto e}(Z) := B^{-\frac{1}{2}} \cdot Z \cdot B^{-\frac{1}{2}};$$
$$\Gamma_{e \mapsto G}(Z) := G^{\frac{1}{2}} \cdot Z \cdot G^{\frac{1}{2}},$$

where $G \in \mathcal{S}_{++}^n$ is a learnable biasing parameter that is optimized through matrix backpropagation during training. To manage computational overhead per minibatch, Brooks et al. (2019) performed only 1 step of the iterative update algorithm to solve (5). During testing, the layer uses the dataset's estimated Fréchet mean, computed via exponential smoothing of the batch means.

To mitigate potential limitations due to crude Fréchet mean estimates Kobler et al. (2022a) proposed an iterative update scheme, denoted SPDMBN, that can control the error to the latent data's changing

---

[4] Letter $Z$ is commonly recognized as symbols representing latent representations in deep learning.

Fréchet mean during matrix backpropagation training and, upon convergence of preceding feature extraction layers, converge to the latent data's true Fréchet mean. They also showed that if the biasing parameter is kept fixed to the identify matrix ($G = e$), the combination of SPDMBN and LOG layers can learn to transport the SPD data to vary around $e$ and project the data to the tangent space $\mathcal{T}_e \mathcal{S}_{++}^n$

Different from Riemannian BN, SPDMBN can also learn to re-scale the SPD data via keeping track of the dataset's Fréchet variance, defined in (6), which aids the matrix backpropagation learning process (Kobler et al., 2022b).

## H    RELAXATION DEVIATION

*Proof.* The $\delta$-width tubular neighborhood of a geodesic $\exp_e(tu)$ on $(\mathcal{S}_{++}^n, g^{AIRM})$ always exists inside its normal bundle according to general tubular neighborhood theorem (Lang, 2012).

For any predefined unit speed $u \in \mathcal{T}_e \mathcal{S}_{++}^n$, let the subspace of speed vectors $\mathcal{U} := \{tu | t \in \mathbb{R}\} \subset \mathcal{T}_e \mathcal{S}_{++}^n$. We can define the orthogonal complement of $\mathcal{U}$ as $\mathcal{U}^\perp := \{u^\perp | g_e^{AIRM}(u^\perp, u) = 0\} \subset \mathcal{T}_e \mathcal{S}_{++}^n$. Note that the orthogonal complement $\mathcal{U}^\perp$ is not an empty set, as the space of skew-symmetric matrices is the orthogonal complement of the space of symmetric matrices.

For each $i \in \{1, ..., N\}$, suppose $\log_e(S_i)^\perp$ are orthogonal components of $\log_e S_i$ with respect to $u$ as follows,

$$\log_e(S_i)^\perp = \log_e(S_i) - g_e^{AIRM}(\log_e(S_i), u) \cdot u \in \mathcal{U}^\perp.$$

Assuming $\delta \in (0, 1]$ represents the deviation magnitude, it is defined as a magnitude with the direction that is perpendicular to the vector $u$ that passes through $\log_e(S)$ in the tangent space. The proposed approach requires that its projections fall within a $\delta$-width tubular neighborhood around $u$. This is expressed as:

$$u_i^\perp := \delta \cdot \frac{\log_e(S_i)^\perp}{\|\log_e(S_i)^\perp\|_e} \in \mathcal{U}^\perp, \quad \forall i = 1, ..., N.$$

Since the relaxation parameter $0 \leq \varepsilon < 1$, we measue the cosine value $\cos(\Theta)$ between $u^\perp + g_e^{AIRM}(\log_e(S), u) \cdot u$ and $u$, and get the following inequality:

$$\frac{|g_e^{AIRM}(u_i^\perp + g_e^{AIRM}(\log_e(S_i), u) \cdot u, u)|}{\|u_i^\perp + g_e^{AIRM}(\log_e(S_i), u) \cdot u\|_e} \geq \epsilon, \quad \forall i = 1, ..., N,$$

which implies a formula for $\delta$ as follows,

$$\delta \leq \sqrt{\frac{1 - \varepsilon^2}{\varepsilon^2}} \max_{i = \{1, ..., N\}} g_e^{AIRM}(\log_e(S_i), u).$$

Since $\max_{i=\{1,...,N\}} g_e^{AIRM}(\log_e(S_i), u)$ is upper bounded as it is a finite set, when $\varepsilon \to 1$, we have $\delta \to 0$.

$\square$

## I    COMMON EXPERIMENTAL SETTINGS

In all experiments, we learn the considered architectures from random initializations and optimize the parameters with ADAM (Kingma & Ba, 2017). To propagate gradients through the network standard minibatch-based backprop learning with extension for structure matrices (Ionescu et al., 2015) and manifold-constrained gradients (Absil et al., 2009; Becigneul & Ganea, 2019) is used.

To implement the considered architectures, we either used publicly available python code or implemented the methods in python using the packages torch (Paszke et al., 2019),(Pedregosa et al., 2011) and geoopt (Kochurov et al., 2020).

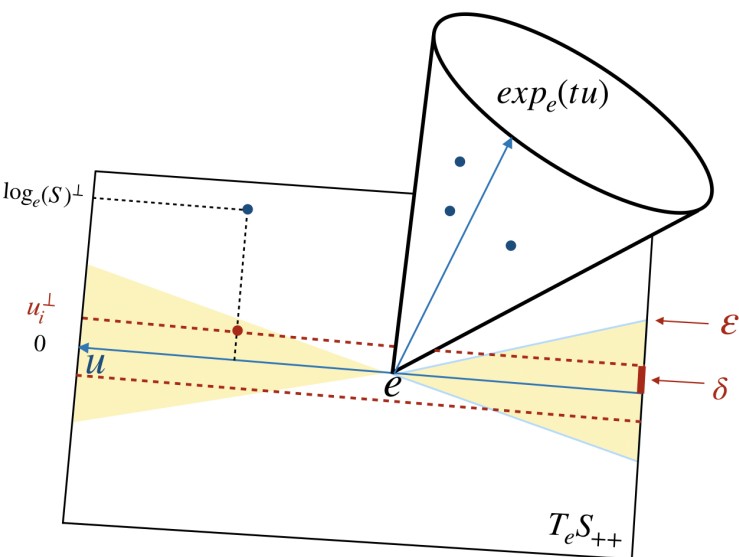

Figure 6: Illustration of Relaxation Deviation: The yellow region represents the double cone, where the maximum cosine value of the included angles is $\varepsilon$. The region within the red dashed lines denotes the tubular neighborhood of width $\delta$ along the speed of geodesic $u$ on the tangent space.

## J   EXPERIMENTAL SETTINGS FOR MOTOR IMAGERY CLASSIFICATION

### J.1   EXPERIMENTAL DATASET

#### J.1.1   KOREAN UNIVERSITY DATASET

The Korean University (KU) dataset, also known as Lee2019MI in MOABB [5], contains EEG signals collected from 54 subjects who participated in a binary-class EEG motor imagery task. The EEG signals were recorded at a sampling rate of 1,000 Hz using 62 electrodes. The EEG signals were segmented from the first second to 3.5 seconds relative to the stimulus onset, resulting in a total duration of 2.5 seconds for each trial. The KU dataset is the largest dataset for the two-movement publicly available motor imagery dataset.

The dataset was partitioned into two sessions, denoted as $S_1$ and $S_2$, each encompassing training and testing phases. In each phase, 100 trials were evenly allocated between right and left-hand imagery tasks. As a result, there were a total of 21,600 trials available for evaluation, where 21,600 trials can be expressed as the product of 2 sessions, 54 subjects, and 200 trials per subject.

#### J.1.2   BNCI2015001 DATASET

The BNCI2015001 dataset from BNCI Horizon 2020 [6] involved 12 participants tasked with performing imagery of right-hand movement versus imagery of movement in both feet. Data were recorded at a sampling rate of 512 Hz and filtered using a bandpass filter ranging from 0.5 to 100 Hz, along with a notch filter at 50 Hz. Recording started at 3.0 seconds following the prompt and continued until the cross period's conclusion at 8.0 seconds, resulting in a total duration of 5 seconds for each trial.

Most of the studies (Subjects $1-8$) were conducted in two sessions, denoted as $S_A$ and $S_B$, which occurred on consecutive days. However, for Subjects 9, 10, 11, and 12, three sessions were denoted as $S_A$, $S_B$, and $S_C$. In each session, every participant completed 100 trials for each class, totaling

---

[5] The MOABB package includes a benchmark dataset designed for motor imagery decoding algorithms, encompassing 12 open-access datasets and involving over 250 subjects, at the following address: `https://github.com/NeuroTechX/moabb`.

[6] The datasets from the BNCI Horizon 2020 project are available for access at the following website `http://bnci-horizon-2020.eu/database/data-sets`.

200 trials per participant per session. Therefore, for self-supervised learning, this dataset can provide a total of 5,600 trials for training. (5,600 trials = 200 trials/session $\times$ 28 sessions).

## J.2 EXPERIMENTAL SCENARIOS

There are two primary evaluation scenarios utilized in this study.

The first scenario involves a 10-fold cross-validation approach. This setup divides each subject's data into 10 equally sized and class-balanced folds. Nine folds are used for training, while one is reserved for testing. This process is repeated ten times using a different fold for testing.

The second scenario is referred to as the holdout scenario. We employ the data from the first session for training, the first 100 trials of the second session for validation, and the remaining 100 trials for testing. It's important to note that the EEG data for these two sessions is typically collected on different days, introducing potential variability.

## J.3 IMPLEMENT DETAILS

### J.3.1 KOREAN UNIVERSITY DATASET

The following two-view DeepGeoSSL model is employed:

- $1^{st}$-View: Two electrodes (C3 and C4);

- $2^{nd}$-View: Twenty electrodes (FC-5/3/1/2/4/6, C-5/3/1/z/2/4/5, and CP-5/3/1/z/2/4/6).

All 21,600 trials are utilized in training the self-supervised learning framework.

For the downstream classification task, we use a tensor-based approach for segmentation in time and frequency domains (Ju & Guan, 2022). In the time domain, there are three segments as follows:

$$0.0 \sim 1.5s, \ 0.5 \sim 2s, \ \text{and} \ 1.0 \sim 2.5s.$$

The frequency band has been partitioned into 9 segments, each spanning a 4Hz bandwidth. These segments cover the range from 4Hz to 40Hz and are evenly distributed within this range. To process the digital signals, Chebyshev Type II filters with 4 Hz intervals were employed. These filters were meticulously designed to ensure a maximum passband loss of 3 dB and a minimum stopband attenuation of 30 dB.

It results in 27 channels (3 temporal segments $\times$ 9 frequency bands). Additionally, the BiMap layer in the architecture takes the input dimension of 20, transforms it to 30, and then returns it to an output dimension of 20. The architecture details can be found in Figure 7.

### J.3.2 BNCI2015001 DATASET

The following two-view DeepGeoSSL model is employed:

- $1^{st}$-View: Two electrodes (FCz and C3);

- $2^{nd}$-View: Twenty electrodes (FC-3/z/4, C-5/3/1/z/2/4/6, and CP-3/z/4).

All 5,600 trials are utilized in training the self-supervised learning framework.

In the time domain, there are three segments as follows:

$$0.0 \sim 1.0s, \ 1.0 \sim 2.0s, \ 2.0 \sim 3.0s, \ 3.0 \sim 4.0s, \ \text{and} \ 4.0 \sim 5.0s.$$

The frequency division was carried out at the same intervals and filter settings as in the KU dataset, resulting in 45 channels (5 temporal segments $\times$ 9 frequency bands). Additionally, the BiMap layer in the architecture takes the input dimension of 13, transforms it to 30, and then returns it to an output dimension of 13. The architecture details can be found in Figure 8.

### J.3.3 GEODESIC SPEED

Since the downstream motor imagery tasks emphasize spectral information for classification, we utilize the largest principal component from Tangent Principle Component Analysis (Tangent PCA) [7] as the initial speed for both $v$ and $w$ as follows,

$$v = w := \text{TPCA}(\{vec(\log_e(\bar{S}_i))\}_{i=1}^N),$$

where $\{\bar{S}_i\}_{i=1}^N$ represents a set of 20-channel EEG signals. In particular, we compute $v = w$ using Tangent PCA for each channel of 27 channels for the KU dataset and 45 channels for the BNCI2015001 dataset.

The choice of this speed can be understood as both modalities projecting signals onto high-information-content spectral principal component directions and keeping the two signals geodesically consistent. This is akin to neural networks augmenting low-information-content signals with sufficient information.

### J.3.4 ESTIMATION OF COVARIANCE MATRICES

It is well-known that the estimation of the sample covariance matrix can be inaccurate, particularly in high-dimensional settings with limited observations. To address this challenge, we employ a regularization technique known as shrinkage, which involves adding a scaled identity matrix (typically multiplied by a small value, such as 1e-2 in this experiment) to the sample covariance matrix.

### J.4 AVERAGE ACCURACY CHANGES

In this experiment, we have two figures that present the average accuracy changes for each participant across six scenarios in two datasets. The baseline is the result obtained by running the 2-channel EEGs using Tensor-CSPNet. The changes are calculated by subtracting the baseline results from DeepGeoCCA ($\varepsilon = 0.05$, $\alpha_1 = 1, \alpha_2 = 1, \alpha_3 = 0.1$ ), as shown in Figures 9 and 10. Supplementary Table 5 additionally summarizes the significance of these differences. In these figures, green areas indicate the subject who has improvements, while blue areas represent those with declines. Our method is effective for a large proportion of subjects in all scenarios.

Table 5: **Simultaneous EEG-fMRI Dataset Results**. Statistical test results for downstream test set accuracy differences in the multi-view EEG experiment, extending Table 2. We used permutation, paired t-tests to identify significant differences between the lower bound (i.e., no pre-training; 2 channel EEG) with DeepGeoCCA for three scenarios ($S_1$, $S_2$, $S_1 \rightarrow S_2$; t-max adjustment for multiple comparisons) and the KU (df=53, 1e4 permutations) and BNCI datasets (df=11, exhaustive permutations). Significant differences are highlighted in bold (p-val $\leq$ 0.05) and trends in italic (p-val $\leq$ 0.1).

| | | KU dataset | | BNCI dataset | |
|---|---|---|---|---|---|
| **scenario** | **SSL pretraining** | mean (std) | t-val (p-val) | mean (std) | t-val (p-val) |
| $S_1$ | ✗ | 62.97 (12.46) | -6.1 (0.0001) | 74.12 (16.50) | -3.1 (0.0391) |
| | DeepGeoCCA | **65.90** (12.95) | - | **77.58** (14.56) | - |
| $S_2$ | ✗ | 65.85 (12.95) | -4.0 (0.0006) | 75.62 (14.58) | -2.9 (0.0469) |
| | DeepGeoCCA | **68.11** (13.64) | - | **78.46** (13.95) | - |
| $S_1 \rightarrow S_2$ | ✗ | 62.89 (12.29) | -4.4 (0.0003) | 75.00 (14.73) | -2.6 (0.0825) |
| | DeepGeoCCA | **65.43** (13.42) | - | *77.83* (13.70) | - |

### J.5 ABLATION STUDY

The loss function of DeepGeoCCA has a total of four hyperparameters: $\varepsilon$ for the $\varepsilon$-geodesic constraints, and three coefficients $\alpha_1$, $\alpha_2$, and $\alpha_3$ for the three loss functions. In the following two ablation studies, we analyze these parameters:

---

[7] We implemented the initialization of $v$ and $w$ using the Tangent PCA approach according to the Geomstats Package in the following address: `https://geomstats.github.io/notebooks/06_practical_methods__riemannian_frechet_mean_and_tangent_pca.html`.

In the first experiment, the overall performance of the model does not vary significantly with different values of $\varepsilon$. Smaller values of $\varepsilon$ lead to a slight improvement in classification performance. The results reported in the main text are based on the setting where $\varepsilon = 0.05$. In the second experiment, we observed that using the variance loss slightly improved the performance compared to not using it. Therefore, in the results reported in the main text, we used the variance loss with a coefficient of $\alpha = 0.1$.

| $\mathcal{L}_\varepsilon$ | $S_1$ | $S_2$ |
|---|---|---|
| Lower Bound | 62.97 (15.72) | 65.85 (16.03) |
| $\varepsilon = 0.9$ | 65.94 (16.10) | 68.19 (16.02) |
| $\varepsilon = 0.7$ | 65.82 (15.35) | 67.77 (16.04) |
| $\varepsilon = 0.5$ | 66.23 (16.10) | 68.06 (16.66) |
| $\varepsilon = 0.3$ | 66.14 (15.93) | 68.09 (16.07) |
| $\varepsilon = 0.1$ | 66.03 (15.56) | 68.13 (15.87) |
| $\varepsilon = 0.05$ | **66.26** (17.05) | **68.27** (17.23) |
| $\varepsilon = 0$ | 66.32 (16.40) | 68.13 (16.55) |

| $\mathcal{L}_\sigma$ | $S_1$ | $S_2$ |
|---|---|---|
| Lower Bound | 62.97 (15.72) | 65.85 (16.03) |
| $\alpha_3 = 0$ | 65.74 (16.59 ) | 67.33 (16.34) |
| $\alpha_3 = 0.1$ | **66.26** (17.05) | **68.27** (17.23) |

(a) $\mathcal{L}_\varepsilon$ in $\mathcal{L}$ ($\alpha_1, \alpha_2 = 1, \alpha_3 = 0.1$).

(b) $\mathcal{L}_\sigma$ in $\mathcal{L}$ ($\alpha_1, \alpha_2 = 1, \varepsilon = 0.05$).

Table 6: Ablation Studies: Average (std across subjects in brackets) classification accuracy for the motor imagery task (higher is better). We considered the KU dataset with scenario 10-fold CV (sessions $S_1$ and $S_2$).

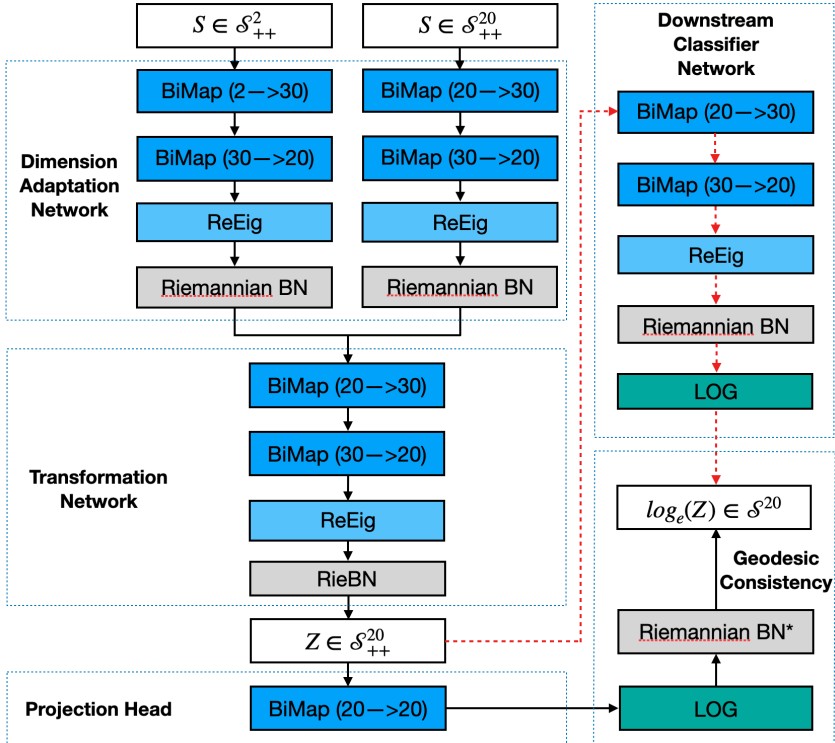

Figure 7: Illustration of the network architecture for the KU dataset. The spatial covariance matrices from both the 2-channel EEG setup (i.e., C3 and C4) and the 20-channel EEG setup (i.e., FC-5/3/1/z/2/4/6, C-5/3/1/z/2/4/5, and CP-5/3/1/z/2/4/6) are transformed into a common dimension using the Dimension Adaptation network. During the training of the self-supervised learning (indicated by the solid black lines), the Transformation network and Projection Head retain the parameters in the neural networks, and geodesic correlation is calculated. The dashed red lines represent the downstream task classifier. Specifically, in the upper right corner of Riemannian BN*, the asterisk $*$ denotes that we use a specially designed Riemannian BN to perform center-parallel translation of outputs to point $e$. Please refer to Section G for information on the layers within the architectures.

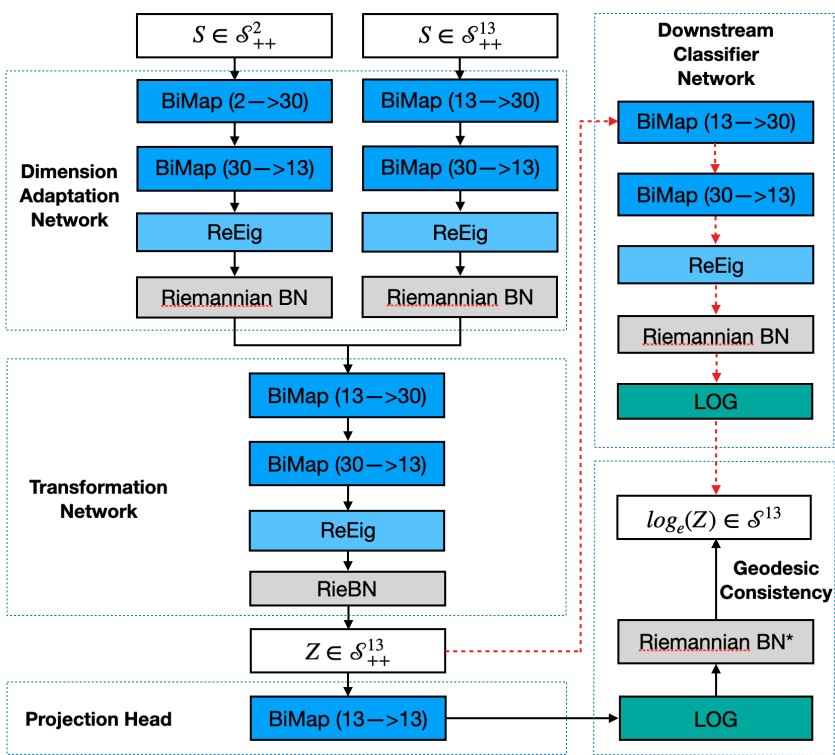

Figure 8: Illustration of the network architecture for the BNCI2015001 dataset. The spatial covariance matrices for the 2-channel EEG setup are derived from FCz and C3, while for the 20-channel EEG setup, they are derived from FC-3/z/4, C-5/3/1/z/2/4/6, and CP-3/z/4. The description of this architecture is consistent with it provided in Figure 7. For a more detailed explanation, please refer to it.

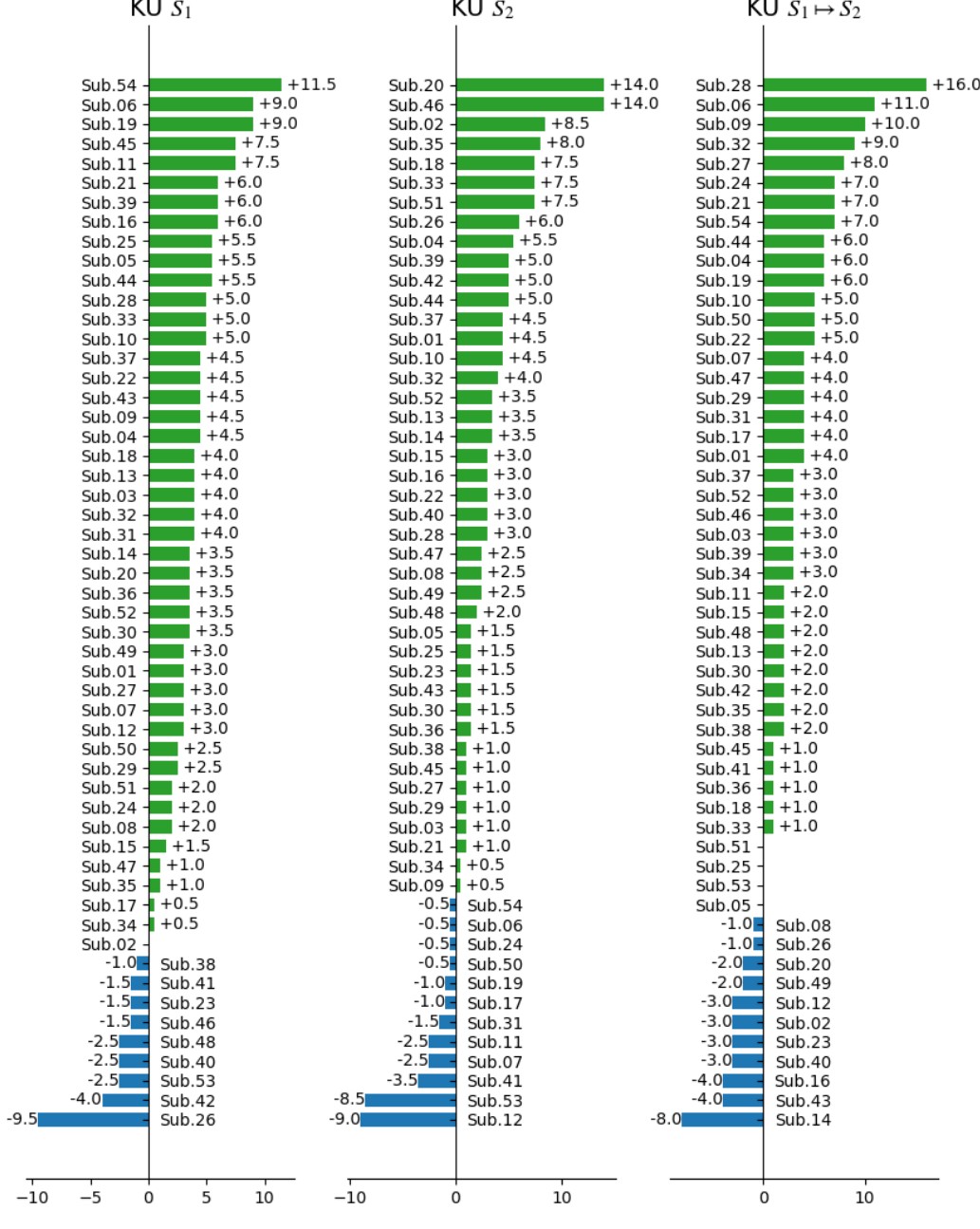

Figure 9: This figure displays the average accuracy changes (%) for each subject in the KU dataset across two 10-fold cross-validation scenarios $S_1$ and $S_2$, and a holdout scenario $S_1 \longmapsto S_2$. Subjects with larger improvements are positioned at the top, arranged in descending order from top to bottom. Green represents an increase, while blue represents a decrease. Corresponding increases are labeled with the respective subject number.

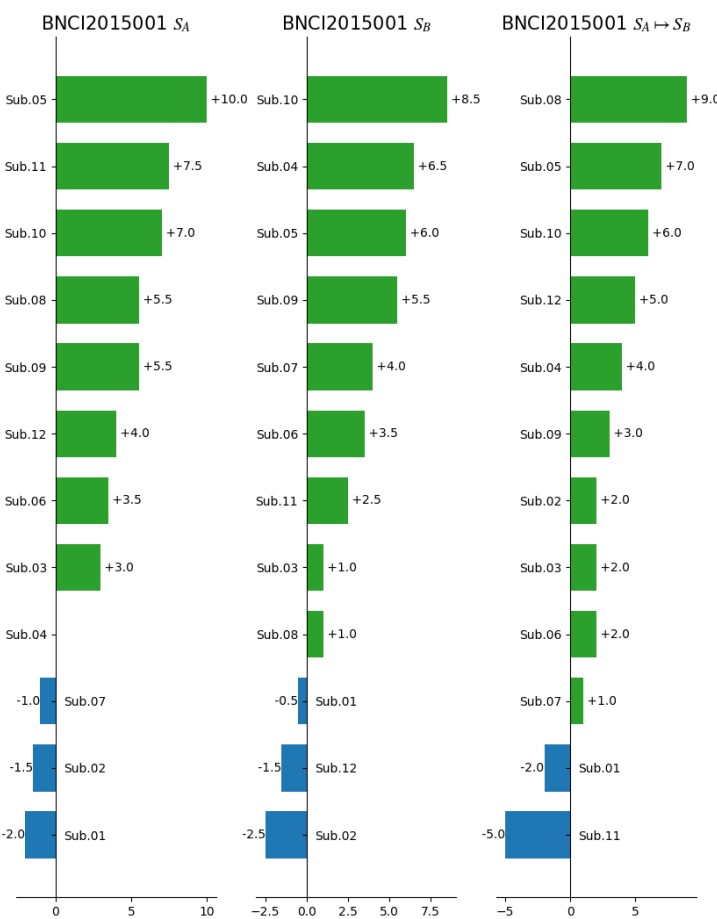

Figure 10: This figure displays the average accuracy changes (%) for each subject in the BNCI2015001 dataset across two 10-fold cross-validation scenarios $S_A$ and $S_B$, and a holdout scenario $S_A \longmapsto S_B$. Subjects with larger improvements are positioned at the top, arranged in descending order from top to bottom. Green represents an increase, while blue represents a decrease. Corresponding increases are labeled with the respective subject number.

## K    SIMULTANEOUS EEG-FMRI DATA

### K.1    EXPERIMENTAL DATASET

We used a publicly available dataset [8] containing simultaneous EEG-fMRI recordings collected from 8 subjects while they were resting in two different conditions (eyes closed or open). This dataset was originally recorded to evaluate EEG artifact correction methods specific to EEG data recorded inside MR scanners (van der Meer et al., 2016b). As a result, it contains several experimental settings. For this analysis, we only consider the simultaneous EEG-fMRI setting where EEG and fMRI data were recorded synchronously. In this setting, the dataset contains 2 runs (i.e., continuous recordings lasting 4.5 minutes) per subject. Within each run, subjects were instructed with visual cues (every 30 seconds) to alternate between the conditions.

For details about the experimental protocol, recording devices and settings, please refer to the accompanying dataset paper (van der Meer et al., 2016a). In brief, EEG data was recorded with 30 MR compatible EEG electrodes placed at standardized locations to cover the whole head. Their electrical activity was sampled at 5 kHz with MR compatible EEG amplifiers. To facilitate EEG artifact correction (Bullock et al., 2021), the EEG amplifiers' clocks were synchronized with the MR scanner. The dataset was recorded inside two different MR scanners (4 subjects inside a Siemens TIM Trio and the rest inside a Siemens Verio). In both MR scanners standard echo-planar imaging (EPI) sequences were used to record fMRI activity. The acquisition parameters were identical except for a slight difference in the repetition time (TR) - 1.95 s for TIM Trio and 2.0 s for Verio. As a result, the fMRI recordings contain volume (i.e., discrete grid with 3 mm isotropic resolution inside the volume covering the subject's head) activity with a sample interval of 2.0 s (or 1.95 s for TIM Trio).

### K.2    DATA PREPROCESSING

The data of each run were preprocessed separately.

**EEG**    data were preprocessed with a custom pipeline. The dataset provides raw and MR gradient-artifact (Allen et al., 2000; Niazy et al., 2005) corrected EEG data; we used the latter. The activity of each EEG channel was bandpass filtered (0.5 to 125 Hz), before attenuating pulse artifacts (van der Meer et al., 2016b). After these EEG-fMRI specific preprocessing steps, we resampled the data at 250 Hz (70 Hz cut-off frequency) and applied further commonly used EEG preprocessing steps. In a nutshell, we detected and interpolated bad EEG channels (Jas et al., 2017), rejected transient single-channel artifacts (de Cheveigné, 2016), re-referenced the channels to their common average, attenuated line-noise at 50 Hz (De Cheveigné, 2020), and rejected transient muscle artifacts (de Cheveigné & Parra, 2014). The residual EEG activity was decomposed into independent components (ICs) (Ablin et al., 2018). These ICs were submitted to an automatized model (Pion-Tonachini et al., 2019) to classify ICs into various artifacts or brain activity. We used tolerant thresholds to maintain on average. 85% (9% std) of the ICs. After rejecting artifact ICs, the residuals were projected back to the original EEG channel space. We excluded 6 temporal and pre-frontal channels, known to capture mostly artifacts, resulting in a total of 24 EEG channels. As a last step, their data was bandpass filtered from 1 to 36 Hz and resampled the data at 125 Hz to extract broad-band brain-activity.

**fMRI**    data preprocessing was similar to (Thomas et al., 2022). First, we used fmriprep (Esteban et al., 2019) to perform standard anatomical and functional preprocessing steps on the raw anatomical and functional scans. In a nutshell, fmriprep performed spatial normalization of fMRI data to a standard space (MNI152NLin2009cAsym), applied slice-time correction and estimated various confounds. As confound regressors, we selected global signal, motion parameters (6 basic parameters) and the first 5 principal components in white matter (WM) and cerebrospinal fluid (SCF). Using the nilearn package [9], we regressed out these confounds from the fMRI data, applied spatial smoothing (Gaussian kernel; FWHM at 3 mm), extracted activity of 128 regions of interest (DiFuMo atlas (Dadi et al., 2020)), applied a bandpass-filter (0.01 Hz to 0.2 Hz), and resampled the output at 1 Hz (i.e., 1 volume per second).

---

[8] https://www.nitrc.org/projects/cwleegfmri_data/
[9] https://nilearn.github.io/

**EEG and fMRI** data were segmented into 8 epochs (20 s duration) using triggers marking the onsets of eyes open and closed conditions. As suggested by Défossez et al. (2023), we used robust z-scoring to standardize the EEG channel and fMRI ROI data per run and clip resulting activity beyond the interval $[-20, 20]$ to reduce effects of gross outliers. The paired data within these epochs were then distributed among train, validation and test sets. We considered 10-fold stratified, leave-one-run-out (LORO) and leave-one-subject-out (LOSO) cross-validation scenarios, as detailed in Figure 11. To boost the number of observations in each set, we extracted sliding windows (10 s duration, 9 s overlap) per epoch.

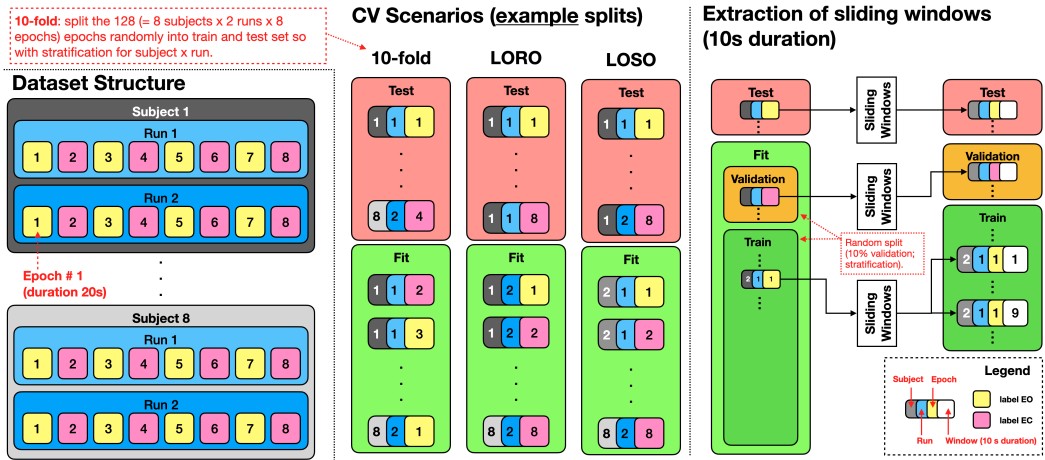

Figure 11: EEG-fMRI dataset. Visualization of the cross-validation (CV) scenarios and sliding windows extraction. (**left**) The considered dataset contains 2 runs for each subject. Each run contains 8 trials with 30 s duration (without breaks in-between) during which the subjects were either resting with eyes open (EO) or eyes closed (EC). We extract 1 epoch per trial (20 s duration, 2 s offset). (**middle**) The epochs were split across CV folds depending on the scenario. We considered 10-fold stratified (10-fold), leave-one-run-out (LORO), and leave-one-subject-out CV. (**right**). After forming the CV splits, sliding windows were extracted from each epoch.

### K.3 IMPLEMENTATION DETAILS

The implementation details provided in this section apply to the simulation and EEG-fMRI experiments.

**Architecture** Figures 12 and 13 visualize the architecture used in the simulation and EEG-fMRI experiments.

**Geodesic Speeds** We fitted the geodesic speeds in data-driven fashion. Specifically, we treated them as learnable parameters on the Stiefel manifold (random initializations). In addition to $\mathcal{L}_\sigma$, we additionally employed them to project the latent tangent space representations $(\log_e(Z_i), \log_e(\bar{Z}_i))$ to scalars and quantify correlation among these in $\mathcal{L}_\rho$.

**Parameter Optimization** We used a minibatch-based training scheme to fit the learnable parameters for 250 epochs. Batches contained 128 paired views (stratified for subject and run). As optimizer, we used ADAM with a learning rate of 1e-3 (5e-3 for the simulations experiment) and applied a weight decay of 1e-3 to all learnable parameters except the manifold-constrained ones. To put more emphasis on maximizing correlation than fulfilling the geodesic projection constraint, we set $\alpha_1 = 1$, $\alpha_2 = 0.25$ and $\alpha_3 = 0$. To discourage the architectures to collapse dimensions, we fixed the learnable rescaling parameter of SPDMBN to 1 so that the Fréchet variance (6) of the latent representations remains approx. constant.

During training, we used a dedicated validation set to monitor the total loss on held out data after every epoch. The parameters that resulted in the lowest validation set loss, were then used to transform the test data.

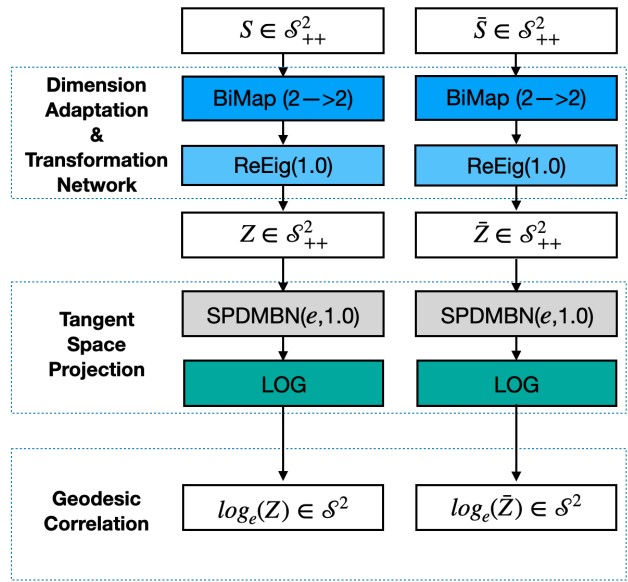

Figure 12: Illustration of Network Architecture for the simulations with paired SPD manifold-valued data. The BiMap layers used unconstrained parameters. For the ReiEig layers, we used a threshold of 1.0 . The tangent space projection module used a SPDMBN layer without learnable parameters, i.e. the rebias parameter was fixed at the identity matrix $e$ and the scaling parameter was fixed to $1$..

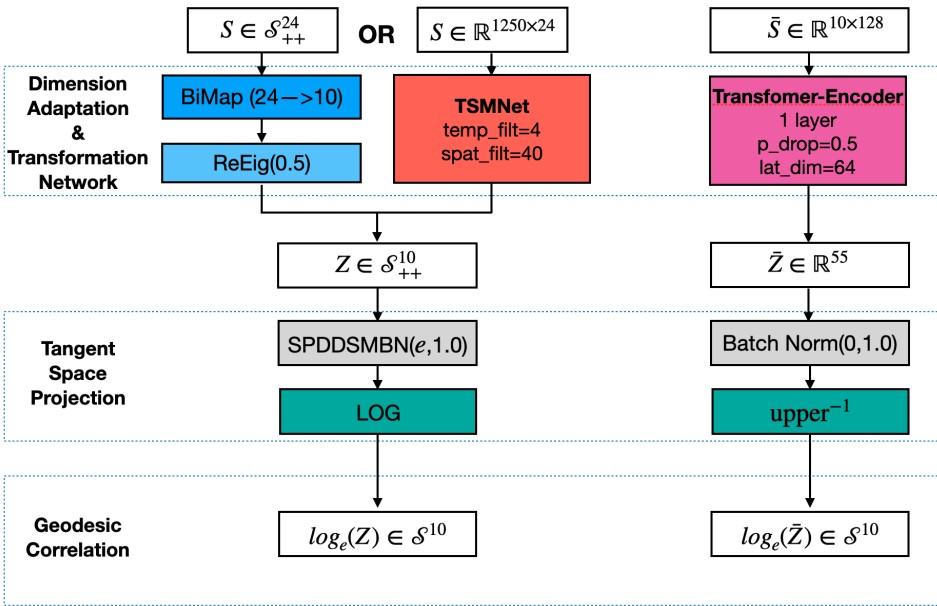

Figure 13: Illustration of Network Architecture for the EEG-fMRI experiment. For EEG views $\bar{S}$, we used either SPDNet or TSMNet. For the ReiEig layer within SPDNet, we used a threshold of 0.5 and unconstrained parameters in BiMap. For TSMNet, we used standard parameters except for 10 latent SPD dimensions instead of 20. The tangent space projection module used a domain-specific SPDMBN layer without learnable parameters. For fMRI views $\bar{S}$, we use a standard a single Transformer-encoder layer as implemented in torch.

### K.4 SUPPLEMENTARY RESULTS

In addition to the 10-fold CV results reported in the main text, this section presents additional results that test generalization across runs (LORO CV scenario) and subjects (LOSO CV scenario). The results are summarized in Table 7 and Figure 14, along with the 10-fold CV results provided in the main text. Comparing the results, we noticed large variability for the LORO and LOSO evaluation scenarios. Particularly for subjects 1 and 5 (highlighted with red dots in Figure 14), all models failed to generalize. To understand why this is the case, we consulted the accompanying publication (van der Meer et al., 2016b) and identified that for these subjects, there was no task effect (eyes open vs. closed) in the EEG data. Since the effect is prominent in the other subjects, naturally, models that utilize the "majority" effect will not generalize to these particular subjects. That is why we report the results in Table 7 with (w/) and without (w/o) test-set folds associated with subjects 1 and 5. Apart from the large variability (mostly due to subjects 1 and 5), we found that the overall effects identified in the 10-fold CV scenario are preserved in the other evaluation scenarios. If we exclude the results for subjects 1 and 5, the average correlation score for all methods increases drastically (approx. 0.2). Particularly, the combination of DeepGeoCCA and TSMNet yields correlations close to 0.7 on held-out data. These additional results clearly demonstrate that our proposed framework can extract representations that generalize to new subjects, and pave the way for potential clinical applications with sufficiently large and diverse datasets.

Table 7: **Simultaneous EEG-fMRI Dataset Results**. Additional test-set results for the EEG-fMRI dataset that extend Table 1 with regard to the considered evaluation scenario. In addition to the 10-fold CV scenario results, reported in Table 1, this table also summarizes results for leave-one-run-out (LORO) and leave-one-subject-out (LOSO) CV scenarios. For LORO and LOSO, we report summary statistics (mean and std) with (w/) and without (w/o) outlier subjects (i.e., subjects 1 and 5).

| EEG model | Evaluation Outlier Method | 10-Fold CV w/ | LORO w/ | LORO w/o | LOSO w/ | LOSO w/o |
|---|---|---|---|---|---|---|
| | | **Correlation ↑** | | | | |
| ✗ | CCA | 0.02 (0.15) | 0.18 (0.35) | 0.34 (0.21) | 0.13 (0.30) | 0.27 (0.17) |
| | RieCCA | 0.38 (0.10) | 0.37 (0.44) | 0.59 (0.13) | 0.24 (0.39) | 0.45 (0.10) |
| Tensor-CSPNet | DeepCCA | 0.42 (0.11) | 0.36 (0.43) | 0.56 (0.23) | 0.30 (0.32) | 0.45 (0.16) |
| | VICReg | 0.29 (0.19) | 0.24 (0.43) | 0.46 (0.21) | 0.31 (0.28) | 0.43 (0.19) |
| | DeepGeoCCA | 0.44 (0.11) | 0.39 (0.45) | 0.58 (0.27) | 0.24 (0.47) | 0.44 (0.35) |
| TSMNet | DeepCCA | 0.29 (0.10) | 0.17 (0.34) | 0.25 (0.32) | 0.12 (0.46) | 0.30 (0.35) |
| | VICReg | 0.38 (0.10) | 0.42 (0.36) | 0.58 (0.22) | 0.25 (0.37) | 0.42 (0.19) |
| | DeepGeoCCA | 0.58 (0.08) | 0.46 (0.45) | 0.68 (0.18) | 0.45 (0.43) | 0.68 (0.14) |
| | | **R2 ↑** | | | | |
| ✗ | CCA | 0.07 (0.08) | -2.32 (1.29) | -2.26 (1.31) | -1.05 (0.70) | -1.19 (0.76) |
| | RieCCA | 0.01 (0.01) | 0.00 (0.02) | -0.00 (0.02) | 0.01 (0.01) | 0.01 (0.01) |
| Tensor-CSPNet | DeepCCA | 0.30 (0.07) | 0.27 (0.13) | 0.28 (0.13) | 0.23 (0.10) | 0.24 (0.12) |
| | VICReg | 0.28 (0.10) | 0.18 (0.12) | 0.20 (0.14) | 0.23 (0.07) | 0.24 (0.07) |
| | DeepGeoCCA | 0.58 (0.02) | 0.56 (0.05) | 0.57 (0.05) | 0.54 (0.07) | 0.54 (0.08) |
| TSMNet | DeepCCA | 0.18 (0.09) | 0.10 (0.06) | 0.11 (0.05) | 0.13 (0.04) | 0.13 (0.05) |
| | VICReg | 0.24 (0.03) | 0.22 (0.11) | 0.23 (0.10) | 0.22 (0.09) | 0.22 (0.09) |
| | DeepGeoCCA | 0.55 (0.02) | 0.53 (0.06) | 0.54 (0.06) | 0.54 (0.02) | 0.54 (0.02) |

## L SPECIFIC CHALLENGES ASSOCIATED WITH NEUROIMAGING DATA

For the neuroimaging tasks that we study, we have to deal with relatively small datasets (recording human neuroimaging data is expensive and time-consuming), non-stationary signals and distribution shifts that limit generalization across days and subjects (also indicated in Figure 4), inherently low signal-to-noise ratios, transient high-variance artifacts (i.e., outliers), and non-linear, poorly understood coupling mechanisms that link views of different modalities (Philiastides et al., 2021). For

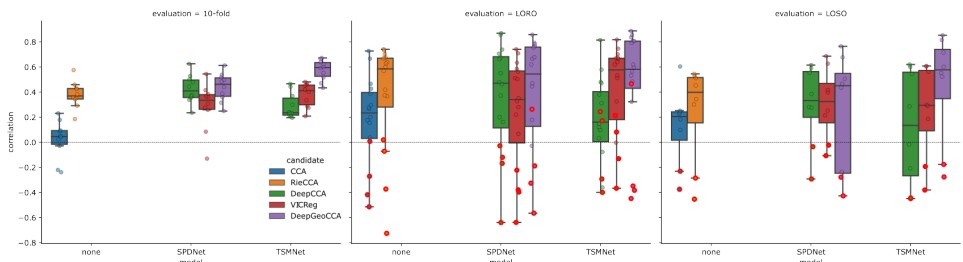

Figure 14: Graphical representation of the results summarized in Table 7 for the correlation metric on test set data. Each dot summarizes a CV split result. (**left**) 10-fold CV scenario. (**middle**) LORO scenario (16 runs=folds). Red dots highlight fold for which the test set was sampled from outlier subjects (subjects 1 and 5). (**right**) LOSO scenario (8 subjects = folds).

more details, let us refer to recent reviews elaborating not only on challenges but also opportunities of EEG BCI (Fairclough & Lotte, 2020) and simultaneous EEG-fMRI (Warbrick, 2022).

