# OpenReview forum: "Deep Geodesic Canonical Correlation Analysis for Covariance-Based Neuroimaging Data"
_ICLR.cc/2024/Conference — ICLR 2024 spotlight_

### Official Review · Reviewer_2nXA · 2023-10-28

**Soundness:** 3 good
**Presentation:** 3 good
**Contribution:** 3 good
**Rating:** 8
**Confidence:** 2

**Summary:**

While researchers began to utilize multi-model data for neuroimaging analysis with the hope to combine distinct information from multiple modalities, the challenge in learning a shared representation space has arisen. The authors pointed out that this is especially nontrivial when dealing with covariance-based data characterized by their geometric structure.

To that end, the authors proposed a metric termed “geodesic correlation” that expands traditional correlation consistency to covariance-based data on symmetric positive definite manifolds and measures the consistency of latent representations obtained from paired views/modalities. They further proposed a deep learning framework termed “DeepGeoCCA” for data on symmetric positive definite manifolds that optimizes the geodesic correlation of unlabeled paired data.

**Strengths:**

1. The motivation to learn a joint latent space with coherence between two latent representations from different modalities is well grounded.

2. Compared to existing methods like DeepCCA and VICReg, both of which learn the correlation relationship between two latent representations, the proposed DeepGeoCCA better retain the geometric relations as analyzed theoretically and empirically.

3. Figures 3 and 4 are well designed.

**Weaknesses:**

1. The authors proposed a metric and a learning framework for "covariance-based data" on symmetric positive definite manifolds, but they have not yet motivated why the EEG/fMRI data are “covariance-based data” and/or why they lie on symmetric positive definite manifolds. While this might be obvious to some people, it is a bit confusing to me.

2. The experiments are a bit weak. While the proposed DeeGeoCCA managed to achieve a higher geodesic correlation, empirical evidence on how this may benefit the learned representation is not clear. Table 2 only showed marginal improvement on the downstream task on relatively small datasets.

**Questions:**

1. Excuse my poor knowledge, but it seems to me that the term “covariance-based data” is not a standard or widely recognized term in the field of statistics or data analysis. Would it be better to define the term somewhere in the paper?

2. On page 4 and 5, you have LaTeX failures (Appendix ??).

3. Possible typo on page 6, “the activity of a latent, shared source, we generated paired views in $S_{++}^2$ $^2$”.

---

> ### Author Response · Authors · 2023-11-16
> **Weakness 1**
>
> We appreciate the reviewers' effort to provide feedback and suggestions. Thank you also for expressing your concerns regarding weaknesses and posing additional questions. Please find our detailed responses below.
>
> > Weakness 1: The authors proposed a metric and a learning framework for "covariance-based data" on symmetric positive definite manifolds, but they have not yet motivated why the EEG/fMRI data are “covariance-based data” and/or why they lie on symmetric positive definite manifolds. While this might be obvious to some people, it is a bit confusing to me.
>
> Our response: Thank you for pointing us to this issue. Based on the feedback of this reviewer, we included a precise motivation for covariance-based data in Appendix A. Specifically, we added motivations for EEG spatial covariance matrices and fMRI functional connectivity matrices as elements on the SPD manifolds.
>
> Firstly, let us formally define covariance-based neuroimaging data as follows:
>
> **Definition (Covariance-Based Neuroimaging Data)**: the covariance matrix of a neuroimaging data segment, represented by $X \in \mathbb{R}^{n_C \times n_T}$, is denoted as $S =  XX^{\top} \in \mathcal{S}_{++}^{n_C}$, where $n_C$ is the number of spatial dimensions and $n_T$ is the signal timesteps.
>
> By definition, covariance-based data is inherently symmetric and positive definite, existing on the Symmetric Positive Definite (SPD) manifolds. Even for degenerate cases (e.g., co-linear spatial dimension), the positive definiteness can be ensured via shrinkage regularization or dimensionality reduction.
>
> Next, let us briefly explain why we chose the second-order statistics of these neuroimaging data as our primary focus for investigation. In a nutshell, EEG spatial covariance matrices capture relevant aspects of neural activity within short to long-time scales. Riemannian geometry-aware methods operating with these covariance-based data are considered state-of-the-art in various EEG application domains, including clinical applications [1,2] and recent brain-computer interface competitions [3,4]. Concerning fMRI, covariance (or correlation) matrices between regions of interest reflect static and dynamic communication between brain networks and their alterations in tasks [5] and pathological conditions [6].
>
> References:
>
> [1] D. Sabbagh, P. Ablin, G. Varoquaux, A. Gramfort, and D. A. Engemann, “Predictive regression modeling with MEG/EEG: from source power to signals and cognitive states,” NeuroImage, vol. 222, p. 116893, Nov. 2020, doi: 10.1016/j.neuroimage.2020.116893.
>
> [2] L. A. W. Gemein et al., “Machine-learning-based diagnostics of EEG pathology,” NeuroImage, vol. 220, p. 117021, Oct. 2020, doi: 10.1016/j.neuroimage.2020.117021.
>
> [3] R. N. Roy et al., “Retrospective on the First Passive Brain-Computer Interface Competition on Cross-Session Workload Estimation,” Front. Neuroergonomics, vol. 3, p. 838342, Apr. 2022, doi: 10.3389/fnrgo.2022.838342.
>
> [4] X. Wei et al., “2021 BEETL Competition: Advancing Transfer Learning for Subject Independence and Heterogenous EEG Data Sets,” in Proceedings of the NeurIPS 2021 Competitions and Demonstrations Track, D. Kiela, M. Ciccone, and B. Caputo, Eds., in Proceedings of Machine Learning Research, vol. 176. PMLR, Dec. 2022, pp. 205–219.
>
> [5] H.-J. Park and K. Friston, “Structural and Functional Brain Networks: From Connections to Cognition,” Science, vol. 342, no. 6158, p. 1238411, Nov. 2013, doi: 10.1126/science.1238411.
>
> [6] A. Yamashita et al., “Generalizable brain network markers of major depressive disorder across multiple imaging sites,” PLOS Biology, vol. 18, no. 12, pp. 1–26, Dec. 2020, doi: 10.1371/journal.pbio.3000966.

---

> > ### Author Response · Authors · 2023-11-16
> > **Weakness 2 (Part 1)**
> >
> > > Weakness 2: The experiments are a bit weak. While the proposed DeeGeoCCA managed to achieve a higher geodesic correlation, empirical evidence on how this may benefit the learned representation is not clear. Table 2 only showed marginal improvement on the downstream task on relatively small datasets.
> >
> > Our response (**Part 1**): This is an excellent point! Thank you for expressing your concern about the experimental results. Apart from a general motivation in the introduction and short sentences in section 4, we provided little context about the specific challenges associated with EEG BCI and simultaneous EEG-fMRI data. The diverse audience reading ICLR contributions can indeed benefit from a brief recap of the main challenges to see how our contribution advances the field. That is why we will emphasize the challenges in the main text and reference a new section in the appendix that elaborates on these challenges. Let us briefly summarize the specific challenges in the next paragraph before we provide further support for the significance of our experimental results in subsequent paragraphs.
> >
> > For the neuroimaging tasks that we study, we have to deal with relatively small datasets (recording human neuroimaging data is expensive and time-consuming), non-stationary signals, distribution shifts (severe across subjects; also indicated in Figure 4), low signal-to-noise ratios, and non-linear coupling mechanisms that link views of different modalities. For more details, let us also refer to recent reviews elaborating on the challenges and opportunities of EEG BCI [1] and simultaneous EEG-fMRI [2].
> > Concerning simultaneous EEG-fMRI, the precise coupling between EEG and fMRI dynamics is actually not well understood [3]. As stated in the submitted manuscript (section 4.2), previous data-driven approaches employed extensive domain knowledge to fuse highly pre-processed EEG and fMRI representations or tried to transcode them [4]. Our approach is fundamentally different. DeepGeoCCA operates with minimally pre-processed data and lets the model learn to identify representations that fuse common information in EEG and fMRI. This methodological advancement opens the door for new insight into their coupling mechanism and large-scale brain dynamics. Beyond the potential impact in neuroscience, our results also form the basis for clinical applications. In this context, the learned representations have to be generalized to unseen subjects. Knowing that cross-subject transfer is a major challenge, reviewer Mrh1 asked us to test cross-subject generalization with a leave-one-subject-out (LOSO) cross-validation scheme. The additional results (summarized in our response to weakness 2 of reviewer Mrh1) clearly demonstrate that DeepGeoCCA can learn stable representations whose dynamics generalize to new subjects, thereby creating the basis for potential future applications in clinical practice. This new result poses an excellent example of empirical evidence on how our framework benefits the learned representations.
> >
> > Motivated by the emergence of mobile, low-cost EEG devices with few channels, we selected the multi-view EEG BCI experiment to showcase the effectiveness of DeepGeoCCA in downstream tasks. We find that pre-training the considered model with DeepGeoCCA yields a statistically significant improvement upon the lower bound in both datasets and considered evaluation scenarios (Table 2 and Rebuttal Table 2), demonstrating our proposed approach's utility. Beyond this, we find that VicReg and DeepCCA also prove useful to fit the considered geometric deep learning model. Note that the combination of the geometric deep learning model and the task can be considered novel in its own right.
> >
> > In summary, it is precisely because our target neuroimaging tasks involve small datasets, non-stationary signals, low signal-to-noise ratios, and challenging non-standard image structures that inspired the design of our geometric deep learning framework. This also convinces us that our current approach is highly competitive for these neuroimaging tasks.

---

> ### Author Response · Authors · 2023-11-16
> **Weakness 2 (Part 2)**
>
> Our response (**Part 2**):
>
> **Rebuttal Table 2**: Auxiliary multi-view EEG downstream results summarize the statistical results in Table 2 in the manuscript. We used permutation, paired t-tests to identify significant differences between the lower-bound (i.e., no pre-training) with DeepGeoCCA for three scenarios ($S_1$, $S_2$, $S_1 \rightarrow S_2$; t-max adjustment for multiple comparisons) and the KU (df=53, 1e4 permutations) and BNCI datasets (df=11, exhaustive permutations).
>
> |                       | dataset         |         KU         |               |        BNCI        |               |
> |-----------------------|-----------------|:------------------:|:-------------:|:------------------:|:-------------:|
> |                       |                 | accuracymean (std) | t-val (p-val) | accuracymean (std) | t-val (p-val) |
> |              scenario | SSL pretraining |                    |               |                    |               |
> |                 $S_1$ |              no |      62.97 (12.46) | -6.1 (0.0001) |      74.12 (16.50) |  -3.1 (0.039) |
> |                       |      DeepGeoCCA |      65.90 (12.95) |             - |      77.58 (14.56) |             - |
> |  $S_2$                |              no |      65.85 (12.95) | -4.0 (0.0001) |      75.62 (14.58) |  -2.9 (0.047) |
> |                       |      DeepGeoCCA |      68.11 (13.64) |             - |      78.46 (13.95) |             - |
> | $S_1 \rightarrow S_2$ |              no |      62.89 (12.29) | -4.4 (0.0001) |      75.00 (14.73) |  -2.6 (0.083) |
> |                       |      DeepGeoCCA |      65.43 (13.42) |             - |      77.83 (13.70) |             - |
>
>
> References:
>
> [1] S. H. Fairclough and F. Lotte, “Grand Challenges in Neurotechnology and System Neuroergonomics,” Front. Neuroergonomics, vol. 1, p. 602504, Nov. 2020, doi: 10.3389/fnrgo.2020.602504.
>
> [2] T. Warbrick, “Simultaneous EEG-fMRI: What Have We Learned and What Does the Future Hold?,” Sensors, vol. 22, no. 6, p. 2262, Mar. 2022, doi: 10.3390/s22062262.
>
> [3] M. G. Philiastides, T. Tu, and P. Sajda, “Inferring Macroscale Brain Dynamics via Fusion of Simultaneous EEG-fMRI,” Annu. Rev. Neurosci., vol. 44, no. 1, pp. 315–334, Jul. 2021, doi: 10.1146/annurev-neuro-100220-093239.
>
> [4] X. Liu, T. Tu, and P. Sajda, “Inferring latent neural sources via deep transcoding of simultaneously acquired EEG and fMRI.” arXiv, Nov. 2022. http://arxiv.org/abs/2212.02226

---

> > ### Author Response · Authors · 2023-11-16
> > **Question 1,2 and 3**
> >
> > > Q1: Excuse my poor knowledge, but it seems to me that the term “covariance-based data” is not a standard or widely recognized term in the field of statistics or data analysis. Would it be better to define the term somewhere in the paper?
> >
> > Our response: This is a great suggestion, and we added a definition and motivation for interested readers in Appendix A! For people in different fields, especially scholars in traditional statistics and data science, the terminology of “covariance-based data” does introduce some ambiguity. Since we feel that this question is related to weakness 1, please refer to our response to weakness 1 and provide additional motivating examples below.
> >
> > Covariance-based data is encountered in many neuroimaging domains. For example, in resting-state functional magnetic resonance imaging (rs-fMRI), functional connectivity among various brain regions is studied via estimating and decomposing covariance (or correlation) matrices. The authors in [1] use "covariance-based" to emphasize and distinguish between two types of functional connectivity metrics, namely covariance-based functional connectivity and correlation-based functional connectivity.
> > In addition to EEG in neuroimaging data, Magnetoencephalography (MEG) spatial covariance can similarly characterize the reciprocal interconnections between brain regions and has been employed in studies related to correlation decoding. Hence, in another work [2], the authors use "covariance-based (connectivity) decoding" to refer to the MEG signal decoding based on covariance-based functional connectivity.
> >
> > Reference:
> >
> > [1]. Strain, Jeremy F., et al. "Covariance-based vs. correlation-based functional connectivity dissociates healthy aging from Alzheimer disease." Neuroimage 261 (2022): 119511. https://www.sciencedirect.com/science/article/pii/S1053811922006279
> >
> > [2]. Mantegna, Francesco, et al. "Covariance-based Decoding Reveals Content-specific Feature Integration and Top-down Processing for Imagined Faces versus Places." bioRxiv (2022): 2022-09. https://www.biorxiv.org/content/10.1101/2022.09.26.509536v1.full
> >
> > > Q2: On page 4 and 5, you have LaTeX failures (Appendix ??).
> >
> > Our response: We are deeply sorry for the oversight, which resulted from separating the main and supplementary parts at that time. We rectify the latex failures in the revised version. The first “Appendix ??” refers to Section G in the Supplementary material. The second “Appendix ??” refers to Section H in the Supplementary material.
> >
> > > Q3: Possible typo on page 6, “the activity of a latent, shared source, we generated paired views in $\mathcal{S}_{++}$.
> >
> > Thank you for reporting this typo back to us. We apologize for introducing this and other typos in the submitted manuscript. In the meantime, we proofread extensively and improved the revised manuscript's spelling and grammar.

---

> ### Comment · Reviewer_2nXA · 2023-11-23
> **Response to Rebuttal**
>
> I would like to thank the authors for providing the highly detailed rebuttal. I have increased the rating from 6 to 8. Good job on the explanations and extra work.

---

> > ### Author Response · Authors · 2023-11-23
> >
> > To Reviewer 2nXA,
> >
> > We extend our sincere gratitude for not only appreciating this work and enhancing the rating but also for your valuable insights and suggestions. We are truly honored and delighted to witness the substantial improvement of this article based on your thoughtful feedback.
> >
> > Sincerely,
> >
> > Submission4904 Authors

---

### Official Review · Reviewer_7bLk · 2023-10-31

**Soundness:** 3 good
**Presentation:** 3 good
**Contribution:** 3 good
**Rating:** 6
**Confidence:** 1

**Summary:**

This paper introduces a novel measure called geodesic correlation and proposes an innovative geometric deep learning framework called DeepGeodesic Canonical Correlation Analysis (DeepGeoCCA) to enhance the consistency of latent representations obtained from paired views in covariance-based neuroimaging data. The authors demonstrate the effectiveness of DeepGeoCCA on both EEG-fMRI and fMRI-fMRI datasets, achieving state-of-the-art performance on several benchmark tasks. The paper's contributions include the introduction of geodesic correlation, the development of DeepGeoCCA, and the demonstration of its effectiveness on various neuroimaging datasets.

**Strengths:**

The paper presents a novel measure called geodesic correlation and proposes an innovative geometric deep learning framework called DeepGeoCCA to enhance the consistency of latent representations obtained from paired views in covariance-based neuroimaging data. The paper is well-written and clearly explains the concepts and methods used in the study.

Originality: The introduction of geodesic correlation is a novel measure that expands traditional correlation consistency to covariance-based data on the symmetric positive definite (SPD) manifold. The proposed DeepGeoCCA framework is also highly original, as it is the first completely data-driven approach to learn latent representations from paired EEG-fMRI data. The effectiveness of DeepGeoCCA is demonstrated on both EEG-fMRI and fMRI-fMRI datasets, achieving state-of-the-art performance on several benchmark tasks.

Quality: The paper is clearly presented and well-structured. With detailed descriptions of the datasets, experimental setups, and evaluation metrics used in the study, it is easy for other researchers to reproduce their results.

Clarity: The paper is well-written and the figures are clearly plotted to illustrate the concepts and methods used in the study.

Significance: The paper is highly significant, as it presents a novel measure and framework for enhancing the consistency of latent representations obtained from paired views in covariance-based neuroimaging data. The proposed DeepGeoCCA framework has the potential to improve our understanding of the dynamic relationships among brain networks and their alterations in pathology, which could lead to improved diagnosis and treatment of neurological disorders. The framework is also not limited to neuroimaging data and can be applied to other types of data with geometric structures, making it a valuable contribution to the field of deep learning.

**Weaknesses:**

The clarity of presentation needs to be improved, with typos scattered throughout the paper:
In Section 2.1, I would suggest using the notation $S^n$ (instead of Sym_n) to denote the space of n × n real symmetric matrices, to be consistent with the the notation S^n_++ for the space of real n × n symmetric and positive definite matrices.

"For a more comprehensive proof, please consult Appendix ??."
"In addition, we have use(d) the relaxed"
Section 3.3, in the definition of $L_epsilon$ formula, there is a redundant right parenthesis ")".

**Questions:**

The authors demonstrate the effectiveness of DeepGeoCCA on both EEG-fMRI and multiview EEG datasets, but it is not clear how the proposed framework would perform on other types of neuroimaging data, such as diffusion tensor imaging (DTI) or positron emission tomography (PET) data. Can the authors comment on the potential applicability of DeepGeoCCA to these types of data?

---

> ### Author Response · Authors · 2023-11-16
> **Both Weakness and Question**
>
> Let us start by thanking the reviewer for his/her effort to assess our submission and the provided feedback. We enjoyed reading the concise summary and strength statements, which accurately highlight all our paper’s contributions. Please find below our responses to your concerns about weaknesses and additional questions.
>
>
> > Weakness: The clarity of presentation needs to be improved, with typos scattered throughout the paper: In Section 2.1, I would suggest using the notation (instead of Sym_n) to denote the space of n × n real symmetric matrices, to be consistent with the the notation S^n_++ for the space of real n × n symmetric and positive definite matrices. "For a more comprehensive proof, please consult Appendix ??." "In addition, we have use(d) the relaxed" Section 3.3, in the definition of
>  formula, there is a redundant right parenthesis ")."
>
> Our response:  We are really sorry for submitting a manuscript with many typos. At the time of submission, we had to realize that we did not reserve enough time for editing and proofreading our submission.
> Thank you very much for reporting these typos back to us. In the meantime, we proofread extensively and improved spelling and grammar in the revised manuscript.
>
>
> > Question: The authors demonstrate the effectiveness of DeepGeoCCA on both EEG-fMRI and multiview EEG datasets, but it is not clear how the proposed framework would perform on other types of neuroimaging data, such as diffusion tensor imaging (DTI) or positron emission tomography (PET) data. Can the authors comment on the potential applicability of DeepGeoCCA to these types of data?
>
> Our response: Absolutely! The framework proposed in this study can be applied to any paired views where one or both are SPD matrices, and the goal is to maximize geodesic correlation among their latent representations.
>
> For other modalities of neuroimaging data, such as diffusion tensor imaging (DTI) or positron emission tomography (PET) data, we find it a very interesting exploration. While we are not experts in DTI and PET ourselves, through a literature search, we have found that many related studies have attempted to leverage the covariance matrix of DTI, making it a highly relevant potential application for our framework. For example, a three-dimensional diffusion tensor represents the covariance of the Brownian motion of water within a voxel and is, therefore, mandated to be a symmetric, positive-definite matrix. Therefore, DTI has been formulated in the SPD manifold and explored using geometric statistics as early as 2007 by P. Thomas Fletcher and Sarang Joshi [1].
>
> It is essential to note that the two modalities must be "paired.” In our simultaneous EEG-fMRI task, the alignment is based on simultaneously collected timestamps, i.e., temporal dynamics. In our two-view EEG task, they are also aligned based on timestamps. For novel applications with DTI or PET data, it is necessary to identify tasks with paired views. In a clinical setting, the task could be disease diagnosis. Naturally, views associated with the same subject could be considered as paired. Consequently, the representations learned with our framework might even impact downstream, clinically relevant tasks in related domains.
>
> Reference:
>
> [1]. Fletcher, P. Thomas, and Sarang Joshi. "Riemannian geometry for the statistical analysis of diffusion tensor data." Signal Processing 87.2 (2007): 250-262.

---

### Official Review · Reviewer_Mrh1 · 2023-10-31

**Soundness:** 3 good
**Presentation:** 3 good
**Contribution:** 2 fair
**Rating:** 8
**Confidence:** 4

**Summary:**

The manuscript presents a CCA method, called DeepGeoCCA, that can be applied to SPD-valued data.  The target application is to relate the covariance matrices (encoding brain connectome) from multiple modalities of neuroimaging data. The method first maps the covariance matrices from two-domains (may be of different sizes, e.g., from EEG and fMRI) to a common SPD cone, where two geodesics are found such that the projections onto the geodesics achieve maximum correlation in the tangent space.

The method was evaluated on a synthetic dataset and on a real dataset of paired EEG-fMRI data. Results show that the method achieved the highest correlation in a cross-validation setting compared to similar non-linear CCA methods.

**Strengths:**

1. CCA analysis on connectome data, brain connectivity matrices, has a great potential in advancing neuroimaging research. The target application can be very broad, e.g., to relate fMRI with structural connectome derived from Diffusion Tensor Imaging.

2. The method can relate covariance matrices of different sizes, which increases the flexibility and can be applied to even broader scenarios, e.g., relating connectome data from different brain parcellations.

**Weaknesses:**

1. The motivation of using the geodesic relaxation (section 3.2) in intuitive a CCA setting. Why do the data points have to lie near the geodesics? In normal CCA or even deep CCA, we never have such constraints to encourage data points to align with the components. Component analysis is about finding directions that explain the data, not the other way around.

2. My biggest concern is the cross-validation setting. These are small datasets (8, 54, and 12 subjects in three datasets). How do you perform 10-fold CV on 8 subjects? I would then assume that the data split is not on the subject level, but on the matrix level (each subject has many covariance matrices generated by a sliding window). These matrices are derived from overlapping time series and have high dependency. Having subject-level split is essential to make the results clinically relevant, otherwise one would just need to find a method that can overfit to the training data.

**Questions:**

1. Is $e$ the identify or the Frechet mean? The text suggests both.

2. There are two "Appendix ??"

3. What is the definition of "double cone"?

4. How do we determine the $\alpha$ weighting parameters?

5. "DeepGeoCCA shared the highest correlation score with Deep-CCA among the deep learning methods (left panel) at the cost of small R2 (right panel)" I thought high correlation automatically means high R2?

6. 10s sliding window is really short for a 2s TR. Are there references for this choice?

---

> ### Author Response · Authors · 2023-11-16
> **Weakness 1 (Part 1)**
>
> We thank the reviewer for taking the time to provide a detailed review of our contribution. We appreciate the on-point summary and the thorough feedback. Thank you also for your account of our contributions and strengths, introducing additional application domains we did not consider before. Please find our detailed responses to the weaknesses and questions below.
>
>
> > Weakness 1: The motivation of using the geodesic relaxation (section 3.2) in intuitive a CCA setting. Why do the data points have to lie near the geodesics? In normal CCA or even deep CCA, we never have such constraints to encourage data points to align with the components. Component analysis is about finding directions that explain the data, not the other way around.
>
> (Part 1): This is a fantastic concern. We apologize for failing to fully articulate some conceptual ideas behind our modeling approach in the submitted manuscript. In this response, we will clarify these thoughts, hoping they contribute to a better understanding of the overall conceptual framework of the paper. Moreover, as requested by this reviewer, we improved the clarity of the revised manuscript by emphasizing key concepts in section 3. Let us first provide some intuitions based on analogies with classical CCA in the next paragraph before we substantiate them in subsequent paragraphs.
>
> Our modeling objective was twofold. Our first objective was to identify non-linear transformations for paired-view data so that the latent SPD representations have maximal geodesic correlation (i.e., similar to the primary CCA objective in linear spaces). Our second objective was to impose additional structure beyond symmetry and positive definiteness, namely, distribution along geodesics, which we promote with the $\varepsilon$-geodesic constraints. We think that the reviewer’s comment is primarily concerned about the second objective (i.e., why the latent representations should be near the geodesics). In our opinion, the second objective extends the linear projection constraint of standard CCA to our problem setting. Linear CCA typically obtains a closed-form solution by transforming the constrained optimization problem into an eigendecomposition problem. We propose a neural network solution to maximize geodesic correlation while using the $\varepsilon$ geodesic constraints to encourage the latent SPD representations to be mostly distributed along the geodesics.
>
> In this study, the introduced approach originates from a fundamental concept within the field of Geometric Statistics, referred to as the geodesic subspace, initially introduced in principal geodesic analysis (PGA) and related works [1-3]. In the context of Riemannian geometry, a geodesic is a curve that locally represents the shortest path between points, acting as a generalization of a straight line. Moreover, the geodesic subspace can be viewed as an expansion of linear spaces from Euclidean to Riemannian geometry. The concept of the geodesic subspace is pivotal in PGA as the projection on this subspace preserves the Riemannian distance, representing the manifold-valued data's variance.
>
> In a formal sense, a submanifold $H$ of manifold $M$ is considered geodesic at the point $p \in H$ if all geodesics of $N$ that pass through $p$ remain geodesic of $M$. A geodesic submanifold $H$ at the Fréchet mean $\mu$ is represented as the exponential mapping of the linear span of $r$ tangent vectors {$w_i$}$\_{i=1}^r$ $\in$ $\mathcal{T}_{\mu} M$
>
> as $N = \exp_{\mu}$ $\big($span({$w_i$}$\_{i=1}^r)$$\big)$.
>
> In PGA, tangent vectors {$w_i$}$\_{i=1}^r$ are computed by sequentially maximizing the Riemannian distance between each data point and the Fréchet mean. This process captures the variance of the data at each step while progressively removing the influence of previously derived directional projections. This completes the manifold-value data's dimensionality reduction. In this context, $w_1$ can be analogous to the first principal component in principal component analysis (PCA). This work goes beyond finding the geodesic submanifold's tangent vectors {$w_i$}$\_{i=1}^r$, obtaining a novel low-dimensional data representation is crucial.
>
> References:
>
> [1]. Fletcher, P. T., and Joshi, S. (2007). Riemannian geometry for the statistical analysis of diffusion tensor data. Signal Processing, 87(2), 250-262.
>
> [2]. Fletcher, P. Thomas, et al. "Principal geodesic analysis for the study of nonlinear statistics of shape." IEEE transactions on medical imaging 23.8 (2004): 995-1005.
>
> [3]. Thomas Fletcher, P. "Geodesic regression and the theory of least squares on Riemannian manifolds." International journal of computer vision 105 (2013): 171-185.

---

> > ### Author Response · Authors · 2023-11-16
> > **Weakness 1 (Part 2)**
> >
> > (Part 2):
> >
> > Canonical Correlation Analysis (CCA) aims to identify the maximum correlation between two variables. Specifically, as outlined in Section 2.2, CCA aims to find linear transformations for paired matrices $X \in \mathbb{R}^{N \times p}$ and $\bar{X} \in \mathbb{R}^{N \times q}$, denoted as $w \in \mathbb{R}^p$ and $\bar{w} \in \mathbb{R}^q$, respectively, to maximize the correlation between these two linear combinations $Xw$ and $\bar{X}\bar{w}$. CCA is typically transformed into an eigenvalue decomposition problem using the Lagrange multipliers method. Consequently, the objective often involves identifying a set of $r$ pairs of canonical variables along with corresponding linear transformations $W\in \mathbb{R}^{p \times r}$ and $\bar{W} \in \mathbb{R}^{q \times r}$, where each column in these matrices signifies a mapping to the respective pair of the canonical covariate. A constraint is imposed to render them uncorrelated to one another to ensure that each pair of canonical variables captures distinct phenomena.
> >
> > When extending the CCA method to Riemannian manifolds, [4] was the first to use geodesic submanifolds as a replacement for linear subspaces. They did not underscore the orthogonal constraints for each pair of canonical covariates and obtained them through a nonlinear optimization approach. Each resulting pair of canonical covariates corresponds to tangent vectors {$w_i$}$\_{i=1}^r$ and {$\bar{w}_i$}$\_{i=1}^r$ for two geodesic submanifolds. **Therefore, data lying on the geodesic can be seen as projecting the data onto a lower-dimensional space by analogy with a linear transformation in classical CCA. Each tangent vector in the tangent space corresponds to a column vector in the linear transformation (canonical covariates).**
> >
> > In this study, our initial target was to enhance the temporal dynamic consistency of simultaneously recorded EEG-fMRI signals. We opt for only one geodesic path for each modality to streamline the method. This design simplification indeed leads to a well-defined convex problem and computational advantages in our deep learning architecture. In particular, opting for only one geodesic path for each modality is equivalent to utilizing the first pair of canonical covariates in CCA. In other words, the tangent vector $w_1$, a symmetric matrix (tangent vector for SPD manifolds), precisely serves as the first canonical covariate. Thus, while the mechanism for obtaining $w_1$ differs from that obtained through eigenvalue decomposition in CCA or DeepCCA, our proposed component analysis method inherently possesses a directional aspect.
> >
> >
> > References:
> >
> > [4]. Kim, Hyunwoo J., et al. "Canonical correlation analysis on Riemannian manifolds and its applications." Computer Vision–ECCV 2014: 13th European Conference, Zurich, Switzerland, September 6-12, 2014, Proceedings, Part II 13. Springer International Publishing, 2014.

---

> > > ### Author Response · Authors · 2023-11-16
> > > **Weakness 2 (Part 1)**
> > >
> > > **Part 1**: Indeed, in the submitted manuscript, we did not analyze the generalization of our framework across subjects - which is the ultimate test for clinical relevance. Let us briefly explain why we initially did not provide subject-level CV results in the next paragraph and detail the employed CV approach before we provide the requested experimental results for the EEG-fMRI experiment.
> > >
> > > For the EEG-EEG experiment (section 4.3 in the manuscript), we used within-session and cross-session evaluation schemes, considered standard evaluation approaches for EEG BCI datasets. We employed a small, public dataset for the simultaneous EEG-fMRI experiment (section 4.2). Knowing that there is typically large variability across subjects and that the dataset only contains data from 8 subjects, we created CV splits that capture this variability. As outlined in Rebuttal Figure 1, the dataset contained alternating trials of eyes open, and eyes closed resting data (30 s duration). These trials were recorded in two runs per subject. We extracted one epoch of 20s duration for each trial with a 5s offset. So, consecutive epochs are separated by 10s gaps, which helps to reduce their dependence. For the 10-fold CV scenario (Rebuttal Figure 1, middle), we employed stratified CV to generate stratified splits for subjects and runs (i.e., each test set contains data from each subject and run). Although the approach is unsuitable for assessing generalization to new subjects or runs, we ensured sliding windows were non-overlapping between epochs. To do so, we extracted epochs, generated CV splits, and finally extracted sliding windows for each epoch (Rebuttal Figure 1, right). We hope that the additional details aid in resolving your concern about the suitability of the employed 10-fold CV scheme.
> > >
> > > [Rebuttal Fig 1](https://anonymous.4open.science/api/repo/iclr24-rebuttal-9BC7/file/fig-eegfmri-cv.png): EEG-fMRI dataset. Visualization of the considered cross-validation scenarios and sliding windows extraction. (left) The dataset contains 2 runs for each subject. Each run contains 8 trials with 30 s duration during which the subjects were either resting with eyes open (EO) or eyes closed (EC). We extract 1 epoch per trial (20 s duration, 2 s offset). (middle) The epochs were split across CV folds depending on the scenario. We considered 10-fold stratified (10-fold), leave-one-run-out (LORO), and leave-one-subject-out CV schemes (right). After forming the CV splits, sliding windows were extracted from each epoch.
> > >
> > > Based on your request to further assess generalization, we conducted 2 additional experiments that test generalization across runs (i.e., leave-one-run-out [LORO] CV) and subjects (i.e., leave-one-subject-out [LOSO] CV). The results for the correlation metric are summarized in Rebuttal Table 1 and Rebuttal Figure 2, along with the 10-fold CV results provided in the initially submitted manuscript. We decided to include these additional results in Appendix K in the supplementary material and refer to them in section 4.2 in the revised manuscript. Comparing the results, we noticed a large variability for the new evaluation schemes. All models failed to generalize, particularly for subjects 1 and 5 (highlighted with red dots in Rebuttal Figure 1). We consulted the accompanying publication to understand why this is the case [1]. We identified no task effect (eyes open vs. closed) in the EEG data for these subjects. Since the effect is prominent in the other subjects, naturally, models that utilize the “majority” effect will not generalize to these particular subjects. That is why we report the results in Rebuttal Table 1 with (w/) and without (w/o) test-set folds associated with subjects 1 and 5. Apart from the large variability (mostly due to subjects 1 and 5), we found that the overall effects we report in the 10-fold CV scheme are preserved in the new evaluation schemes. If we exclude the results for subjects 1 and 5, the average correlation score for all methods increases drastically (approx. 0.2). Particularly, the combination of DeepGeoCCA and TSMNet yields correlations close to 0.7 on held-out data. These additional results clearly demonstrate that our proposed framework has the potential to become a viable clinical tool provided it is fit to sufficiently large and diverse datasets.
> > >
> > > Rebuttal Figure 2 and Rebuttal Table 1 please refer to Weakness (Part 2).
> > >
> > > References:
> > >
> > > [1] J. N. van der Meer et al., “Carbon-wire loop based artifact correction outperforms post-processing EEG/fMRI corrections—A validation of a real-time simultaneous EEG/fMRI correction method,” NeuroImage, vol. 125, pp. 880–894, Jan. 2016, doi: 10.1016/j.neuroimage.2015.10.064.

---

> > > > ### Author Response · Authors · 2023-11-16
> > > > **Weakness 2 (Part 2)**
> > > >
> > > > **Part 2**:
> > > >
> > > > [Rebuttal Fig 2](https://anonymous.4open.science/api/repo/iclr24-rebuttal-9BC7/file/fig-eegfmri-suppl.png): Graphical representation of the results summarized in Rebuttal Table 1. Individual dots represent folds. (left) 10-fold CV scenario. (middle) LORO scenario (16 runs=folds). Red dots highlight outlier subject data (subjects 1 and 5). (right) LOSO scenario (8 subjects = folds).
> > > >
> > > > **Rebuttal Table 1** (Table 7 in Appendix): Additional test-set results for the EEG-fMRI dataset that extend Table 1 in the manuscript with regard to the considered evaluation scheme. In addition to the original 10-fold CV scheme, we tested leave-one-run-out (LORO) and leave-one-subject-out (LOSO) CV. For LORO and LOSO, we report summary statistics (mean and std) with (w/) and without (w/o) subjects 1 and 5.
> > > >
> > > > |                     |                               |     Correlation       mean (sd)    |                    |                    |                    |                    |
> > > > |---------------------|-------------------------------|------------------------------------|--------------------|--------------------|--------------------|--------------------|
> > > > |                     |     CV   scenario             |     10-fold                        |     LORO           |                    |     LOSO           |                    |
> > > > |                     |     subjects       1 and 5    |     w/                             |     w/             |     w/o            |     w/             |     w/o            |
> > > > |     model           |     candidate                 |                                    |                    |                    |                    |                    |
> > > > |     none            |     CCA                       |     0.02 (0.15)                    |     0.18 (0.35)    |     0.34 (0.21)    |     0.13 (0.30)    |     0.27 (0.17)    |
> > > > |                     |     RieCCA                    |     0.38 (0.10)                    |     0.37 (0.44)    |     0.59 (0.13)    |     0.24 (0.39)    |     0.45 (0.10)    |
> > > > |     TensorCSPNet    |     DeepCCA                   |     0.42 (0.11)                    |     0.36 (0.43)    |     0.56 (0.23)    |     0.30 (0.32)    |     0.45 (0.16)    |
> > > > |                     |     VICReg                    |     0.29 (0.19)                    |     0.24 (0.43)    |     0.46 (0.21)    |     0.31 (0.28)    |     0.43 (0.19)    |
> > > > |                     |     DeepGeoCCA                |     0.44 (0.11)                    |     0.39 (0.45)    |     0.58 (0.27)    |     0.24 (0.47)    |     0.44 (0.35)    |
> > > > |     TSMNet          |     DeepCCA                   |     0.29 (0.10)                    |     0.17 (0.34)    |     0.25 (0.32)    |     0.12 (0.46)    |     0.30 (0.35)    |
> > > > |                     |     VICReg                    |     0.38 (0.10)                    |     0.42 (0.36)    |     0.58 (0.22)    |     0.25 (0.37)    |     0.42 (0.19)    |
> > > > |                     |     DeepGeoCCA                |     0.58 (0.08)                    |     0.46 (0.45)    |     0.68 (0.18)    |     0.45 (0.43)    |     0.68 (0.14)    |

---

> > > > > ### Author Response · Authors · 2023-11-16
> > > > > **Question 1,2,3 and 4**
> > > > >
> > > > > > Q1. Is $e$ the identity or the Frechet mean? The text suggests both.
> > > > >
> > > > > Our response: Our response: We always use $e$ to represent the identity element in the text. We employ Riemannian batch normalization layers in the proposed neural network solution with their bias parameter constrained to $e$. Consequently, the data’s Fréchet mean will be transported to $e$. We apologize for any ambiguity and have adjusted section 3 to clarify this issue in the revised version.
> > > > >
> > > > > > Q2. There are two "Appendix ??"
> > > > >
> > > > > Our response: We are deeply sorry for the oversight, which resulted from separating the main and supplementary parts at that time. We rectify this issue in the revised version. The first “Appendix ??” refers to Section G in the Supplementary material. The second “Appendix ??” refers to Section H in the Supplementary material.
> > > > >
> > > > > > Q3. What is the definition of "double cone"?
> > > > >
> > > > > Our response: "Double cone" generally refers to a cone extending infinitely in both directions from the apex; in such cases, it is referred to as a double cone. This study specifically refers to the region formed by relaxing geodesics passing through the identity element (Section 3.2). In Figure 6 (Illustration of Relaxation Deviation) in the Appendix, the yellow region represents a schematic illustration of the double cone formed in the tangent space.
> > > > >
> > > > > > Q4. How do we determine the $\alpha$ weighting parameters?
> > > > >
> > > > > Our response: Our study employs an empirical approach to select parameters suitable for the three sets of experiments. Selecting the loss weighting is not trivial, with the optimal parameter range likely depending on the particular dataset. We adhere to fundamental principles to ensure the robust performance of our approach in practical applications. The weight selection involves balancing various factors, including the magnitudes of numerical values in different modalities, model training stability, convergence speed, and emphasis on each component.
> > > > >
> > > > > Actually, the proposed loss function originates from the Lagrangian for the constrained nonlinear optimization problem introduced in section F of the supplementary material. During that phase, the weights, akin to Lagrange multipliers, could be determined through nonlinear optimization methods. However, due to the incorporation of the relaxation method and the adoption of deep learning architecture, modifications have been applied to the loss function. The first two terms among the three loss components jointly constrain the geodesic correlation, while the third term serves as an additional constraint on variance.
> > > > >
> > > > > Specifically, w.l.o.g., we fixed weight $ \alpha_1 = 1$ and used simulation experiments (similar to section 4.1) to identify meaningful ranges for weight $ \alpha_2 $ while keeping weight $ \alpha_3 = 0$. As expected, the simulation results (not shown) indicate similar effects for changing either $\alpha_2 $ or $ \epsilon $. Therefore, we merely report the submitted manuscript's results for varying $ \epsilon $.
> > > > >
> > > > > In addition, for the EEG-fMRI experiment, we obtained good results with the simulation parameters. So, we kept them fixed. Based on the first results with the multi-view EEG data (section 4.3), we increased $\alpha_2$ and introduced $\alpha_3 > 0 $ to improve convergence and prevent the model from mode collapse in this experiment.
> > > > >
> > > > > In particular, these choices perform well within the considered experiments. In practical applications, optimizing these hyperparameters by tuning them based on the validation set might be helpful.

---

> > > > > > ### Author Response · Authors · 2023-11-16
> > > > > > **Question 5 and 6**
> > > > > >
> > > > > > > Q5. "DeepGeoCCA shared the highest correlation score with Deep-CCA among the deep learning methods (left panel) at the cost of small R2 (right panel)" I thought high correlation automatically means high R2?
> > > > > >
> > > > > > Our response: You are right if correlation and R2 metrics are used to assess the goodness-of-fit for the same quality. In our submission, we employed correlation and R2 to measure different qualities; correlation to assess the primary objective ($ \mathcal{L}$$\_\rho$) and R2 to assess the $\varepsilon$-geodesic constraints ($\mathcal{L}$$\_\varepsilon $).
> > > > > >
> > > > > > Specifically, we used correlation to assess how well the covariate pairs of two modalities (or views) covary and R2 to estimate how much variance each covariate explains within its own latent representation. In the submitted manuscript, we introduced this motivation in section 4.1 (paragraph 2 and footnote 4 on page 6). We feel like clarity could be improved in the revised manuscript. To do so, we placed this part more prominently and highlighted it again in the figure captions to remind the readers that the metrics are used to measure different qualities.
> > > > > >
> > > > > >
> > > > > > > Q6. 10s sliding window is really short for a 2s TR. Are there references for this choice?
> > > > > >
> > > > > > Our response: Yes, 10s windows are indeed rather short. Please note that we resampled the fMRI time series to have a TR of 1 s yielding sliding windows with 10 volumes (see Appendix L.2).  We motivated this choice twofold. First, for the fMRI decoder, we use a similar architecture to [1]. This work considered windows ranging from 10 to 50 TRs (with TRs varying from 0.8 to 2.8 s). Our choice corresponds to their lower limit. Choosing 10 s windows allowed us to extract multiple - yet dependent - observations for each 20 s epoch (see, e.g., Rebuttal Figure 1). Second, the hemodynamic response (HR) in the fMRI signal is typically a non-linear, smoothed, and delayed response to the actual neural activity, peaking at around 4 to 6 s and returning to baseline level within 10 s, followed by an undershoot that decays to baseline within 20 s. Consequently, 10 s windows span most of the HR and should suffice to align fMRI signals with EEG signals, which capture an instantaneous mixture of large-scale brain activity.
> > > > > >
> > > > > > [1] A. Thomas, C. Ré, and R. Poldrack, “Self-Supervised Learning of Brain Dynamics from Broad Neuroimaging Data,” in Advances in Neural Information Processing Systems, 2022, pp. 21255–21269.

---

### Author Response · Authors · 2023-11-16
**To all reviewers and ACs**

We express our gratitude to you for your dedicated time and invaluable feedback, contributing to the recognition of our contributions and the enhancement of our work. Following your feedback, we have made updates to the manuscript.

Here is a summary of two key concerns highlighted by the reviewers:
1. Concerns about theoretical completeness: The reviewer (Mrh1) raised concerns about the necessity of the geodesic constraint.
2. Concerns about the experimental setup: reviewers (Mrh1 and 2nXA) expressed concerns about the rationale and specific parameters used in several experiments in the study.
In addition to these two primary concerns, other issues were noted, including the choice of weights from Mrh1, terminology concerns from 2nXA, exploration of other neuroimaging modalities from 7bLK, and pointing out various typos in the manuscript during the editing process.

We once again express our gratitude to you for providing constructive feedback and insightful concerns. In our responses, we have tried to address these concerns comprehensively from both theoretical analysis and experimental perspectives. Due to time constraints and the limitations of our experience and knowledge, we welcome you to provide new suggestions if our responses are not exhaustive.

As the responses in this round are quite extensive, we have decided to post them earlier to allow more time for your feedback. We have highlighted key points in each response to facilitate you in quickly locating answers corresponding to specific concerns. Moreover, we uploaded a revised manuscript based on the reviewers’ feedback and provided a document highlighting the differences between both versions. The specific revision methods and content for the parts that need modification have all been detailed in this response. Here is the anonymous link to the updated manuscript with all changes highlighted (using latexdiff).

https://anonymous.4open.science/api/repo/iclr24-rebuttal-9BC7/file/231116-main_full-diff.pdf

---

### Meta-Review · Area_Chair_NeZ9 · 2023-12-08

**Metareview:**

The submission proposes deep geodesic canonical correlation analysis and applies it to neuroimaging data.  The method is well motivated and the rebuttal process further clarified the motivation behind the geodesic constraint to the satisfaction of the reviewers.  There was a unanimous recommendation for acceptance of the paper, with reviewers expressing that the rebuttal sufficiently addressed concerns.  In particular, an issue with cross validation was corrected in the rebuttal process to have a leave-one-subject out structure.  This was an important correction to the initially submitted 10-fold CV experiments when there were fewer subjects than folds.  Although that was a big issue, new results provided during the rebuttal were a key improvement that should be integrated into the main text.

**Justification For Why Not Higher Score:**

Variants of non-linear CCA has been applied to neuroimaging data for many years.  The current iteration appears to be a well implemented extension.

**Justification For Why Not Lower Score:**

Unanimous recommendation for acceptance.  Corrected cross-validation in the extra reported experiments.

---

### Decision · Program_Chairs · 2024-01-16

Accept (spotlight)